# Uncoupled Learning Dynamics with $O(\log T)$ Swap Regret in Multiplayer Games

**Ioannis Anagnostides**
Carnegie Mellon University
Pittsburgh, PA 15213
ianagnos@cs.cmu.edu

**Gabriele Farina**
Carnegie Mellon University
Pittsburgh, PA 15213
gfarina@cs.cmu.edu

**Christian Kroer**
Columbia University
New York, NY 10027
christian.kroer@columbia.edu

**Chung-Wei Lee**
University of Southern California
Los Angeles, CA 90007
leechung@@usc.edu

**Haipeng Luo**
University of Southern California
Los Angeles, CA 90007
haipengl@usc.edu

**Tuomas Sandholm**
Carnegie Mellon University
Strategy Robot, Inc.
Optimized Markets, Inc.
Strategic Machine, Inc.
Pittsburgh, PA 15213
sandholm@cs.cmu.edu

## Abstract

In this paper we establish efficient and *uncoupled* learning dynamics so that, when employed by all players in a general-sum multiplayer game, the *swap regret* of each player after $T$ repetitions of the game is bounded by $O(\log T)$, improving over the prior best bounds of $O(\log^4(T))$. At the same time, we guarantee optimal $O(\sqrt{T})$ swap regret in the adversarial regime as well. To obtain these results, our primary contribution is to show that when all players follow our dynamics with a *time-invariant* learning rate, the *second-order path lengths* of the dynamics up to time $T$ are bounded by $O(\log T)$, a fundamental property which could have further implications beyond near-optimally bounding the (swap) regret. Our proposed learning dynamics combine in a novel way *optimistic* regularized learning with the use of *self-concordant barriers*. Further, our analysis is remarkably simple, bypassing the cumbersome framework of higher-order smoothness recently developed by Daskalakis, Fishelson, and Golowich (NeurIPS'21).

## 1   Introduction

Online learning and game theory share an intricately connected history tracing back to the inception of the modern *no-regret framework* with Robinson's analysis of *fictitious play* [Robinson, 1951] and Blackwell's *approachability theorem* [Blackwell, 1956]. Indeed, the no-regret framework addresses the fundamental question of how independent and decentralized agents can "learn" with only limited feedback from their environment, and has led to celebrated connections with game-theoretic equilibrium concepts [Hart and Mas-Colell, 2000, Foster and Vohra, 1997]. One of the remarkable features of these results is that the learning dynamics are fully *uncoupled* [Hart and Mas-Colell, 2000]: each player is completely agnostic to the utilities of the other players. Thus, there

36th Conference on Neural Information Processing Systems (NeurIPS 2022).

is no communication between the players or any centralized authority dictating behavior throughout the game. Instead, the only "coordination device" is the common history of play. An additional desideratum, which is fundamentally tied to the no-regret framework, is what Daskalakis et al. [2011] refer to as *strong uncoupledness*:[1] players have no information whatsoever about the game (even their own utilities), and they only make decisions based on the utilities received as feedback throughout the repeated game.

In this context, it is well-known that there are broad families of no-regret learning algorithms that, after $T$ repetitions, guarantee regret bounded by $O(\sqrt{T})$, and this bound is known to be insuperable in adversarial environments [Cesa-Bianchi and Lugosi, 2006]. However, this begs the question: *What if the player is not facing adversarial utilities, but instead is competing with other learning agents in a repeated game?* This question was first formulated and addressed by Daskalakis et al. [2011], who devised strongly uncoupled dynamics converging with a near-optimal rate of $O(\frac{\log T}{T})$ in zero-sum games, a substantial improvement over the $O(1/\sqrt{T})$ rate obtained via traditional approaches within the no-regret framework. Thereafter, there has been a considerable amount of effort in strengthening their result, leading to extensions along several important lines [Rakhlin and Sridharan, 2013, Syrgkanis et al., 2015, Chen and Peng, 2020, Farina et al., 2019, Daskalakis et al., 2021, Anagnostides et al., 2022a, Wei and Luo, 2018, Foster et al., 2016, Anagnostides et al., 2022b]. In particular, in a recent breakthrough result, Daskalakis et al. [2021] showed that when all players in a general game employ an *optimistic* variant of *multiplicative weights update (MWU)* (henceforth *OMWU*), the *external regret* of each player grows as $O(\log^4(T))$. That result was also subsequently extended to the substantially more challenging performance measure of *swap regret* [Anagnostides et al., 2022a]. Perhaps the main drawback of the latter results is the complexity of the analysis, relying on establishing a refined property for the dynamics they refer to as *higher-order smoothness*. Our primary contribution in this paper is to develop a novel and much simpler framework, which furthermore improves the prior state of the art $O(\log^4(T))$ regret bounds to $O(\log T)$ in general multiplayer games.

## 1.1 Overview of Our Contributions

Before we state our main result, let us first introduce some basic notation. We assume that each player $i \in [\![n]\!]$ selects at every iteration $t$ of the repeated game a probability distribution (mixed strategy) over the set of available actions $\boldsymbol{x}_i^{(t)} \in \Delta(\mathcal{A}_i)$ (see Section 2 for further details). The following theorem is the primary contribution of our work.[2]

**Theorem 1.1** (Precise Statement in Theorem 4.4). *There exist strongly uncoupled no-swap-regret learning dynamics so that when employed by all players with learning rate $\eta = \Theta(1)$, the second-order path lengths of the dynamics up to any time $T \in \mathbb{N}$ are bounded by $O(\log T)$; that is,*

$$\sum_{t=1}^{T} \sum_{i=1}^{n} \|\boldsymbol{x}_i^{(t)} - \boldsymbol{x}_i^{(t-1)}\|_1^2 = O(\log T).$$

We are not aware of even an $o(T)$ bound for the second-order path lengths—under a *time-invariant* learning rate—prior to our work, except for very restricted classes of games such as zero-sum games. The dynamics of Theorem 1.1 combine: (i) the celebrated no-swap-regret template of Blum and Mansour [2007]; (ii) the *optimistic follow the regularizer leader (OFTRL)* algorithm of Syrgkanis et al. [2015]; and (iii) using a *self-concordant barrier* as a regularizer. The latter was introduced in online learning in the seminal work of Abernethy et al. [2008], where the authors obtained the first near-optimal and efficient online learning algorithm for linear bandit optimization; the way we leverage the *log-barrier* in the setting of no-regret learning in games is novel, and crucially leverages the *local norm* induced by the regularizer. The dynamics of Theorem 1.1 are also efficiently implementable (see Remark 4.7).

The implication of Theorem 1.1 is perhaps surprising in view of the inherent *cycling* aspect of no-regret learning in general games. Indeed, it is by now well-understood that any no-regret dynamics

---

[1] Daskalakis et al. [2011] also impose that players are only allowed to (privately) store only a constant number of observed utilities, an assumption also espoused in our work.

[2] For simplicity in the exposition, we use the $O(\cdot)$ notation in our introduction to suppress parameters that depend (polynomially) on the natural parameters of the game; precise statements are given in Section 4.

| Reference | Algorithm | Swap Regret in Games | Adversarial Swap Regret |
|---|---|---|---|
| Blum and Mansour [2007] | *E.g.*, BM-MWU | —— | $O(\sqrt{m\log(m)T})$ |
| Chen and Peng [2020] | BM-OMWU | $O(\sqrt{n}(m\log(m))^{3/4}T^{1/4})$ | $\widetilde{O}(\sqrt{mT})$ |
| Anagnostides et al. [2022a] | BM-OMWU | $O(nm^4\log(m)\log^4(T))$ | —— |
| **This paper** | BM-OFTRL-LogBar | $O(nm^{5/2}\log(T))$ | $O(\sqrt{m\log(m)T})$ |

Table 1: Prior results regarding the no-swap-regret algorithm of Blum and Mansour [2007] (BM). The second column indicates the algorithm internally employed by the "master" BM algorithm; our construction uses OFTRL with log-barrier regularization (Section 3). Further, $m$ is the maximum number of actions available to each player. We point out that in the adversarial swap regret bound we have suppressed lower order factors in terms of $T$. We further remark that the near-optimal *internal regret* guarantee of Anagnostides et al. [2022a] in turn implies $O(nm\log(m)\log^4(T))$ swap regret for each individual player, but is obtained via the algorithm of Stoltz and Lugosi [2005].

will fail to converge—at least for certain games (*e.g.*, see [Milionis et al., 2022]). Nevertheless, Theorem 1.1 implies that players will change their strategies *arbitrarily slowly* as the game progresses. As such, players will observe utilities that exhibit very small variation over time, immediately implying near-optimal swap regret.

**Corollary 1.2** (Precise Statement in Corollaries 4.5 and 4.6). *There exist strongly uncoupled no-swap-regret learning dynamics so that when employed by all players, the individual swap regret of each player is bounded by $O(\log T)$. At the same time, when faced against adversarial utilities each player guarantees $O(\sqrt{T})$ swap regret.*

Corollary 1.2 improves over the prior best bounds of $O(\log^4(T))$ [Daskalakis et al., 2021, Anagnostides et al., 2022a]; a comparison with prior works regarding the algorithm of Blum and Mansour [2007] is given in Table 1. In fact, Corollary 1.2 yields, to our knowledge, the first no-regret guarantee in general games for uncoupled methods when players use a *time-invariant* learning rate, a feature that has been extensively motivated in prior works (see, *e.g.*, the discussion in [Bailey and Piliouras, 2019]). Corollary 1.2 also establishes near-optimality in the adversarial regime as well, a crucial desideratum in this line of work. Finally, swap regret is a powerful notion of hindsight rationality, trivially subsuming external regret. In particular, in light of well-established connections (see Theorem 2.3), we obtain the best known rate of convergence of $O(\frac{\log T}{T})$ to *correlated equilibria* in general games.

**Corollary 1.3.** *There exist strongly uncoupled learning dynamics so that, when employed by all players, the average correlated distribution of play after $T$ repetitions of the game is an $O(\frac{\log T}{T})$-approximate correlated equilibrium.*

From a technical standpoint, our approach is conceptually remarkably simple and direct. Specifically, Theorem 1.1 is shown by first establishing the RVU bound—a fundamental property first identified in [Syrgkanis et al., 2015, Definition 3]—for *swap regret* in Theorem 4.3; the key ingredient is Lemma 4.2, which crucially leverages the local norm induced by the log-barrier regularizer over the simplex. Next, Theorem 1.1 follows directly by making a seemingly trivial observation: *swap regret is always nonnegative*. A related approach was recently employed in [Anagnostides et al., 2022c] for external regret, but only works for very restricted classes of games such as zero-sum. As such, we bypasses the cumbersome framework of higher-order smoothness recently introduced by Daskalakis et al. [2021].

## 1.2 Further Related Work

The first accelerated dynamics in general games were established by Syrgkanis et al. [2015]. In particular, they identified a broad class of no-regret learning dynamics—satisfying the so-called RVU property—for which the *sum* of the players' regrets is $O(1)$. On the other hand, they only obtained an $O(T^{1/4})$ bound for the individual external regret of each player. This is crucial given that the rate of convergence to *coarse* correlated equilibria is driven by the *maximum* of the external regrets. It is important to note that a bound for the sum of the external regrets does not necessarily translate to a bound for the maximum since *external regrets* can be negative. This is in stark contrast to swap regret

(Observation 2.1), a property crucially leveraged in our work. Furthermore, the $O(T^{1/4})$ bounds for the individual external regret in [Syrgkanis et al., 2015] were only recently extended to swap regret by Chen and Peng [2020]. The main challenge with swap regret—which is also the main focus of our paper—is that the underlying dynamics are much more complex, maintaining and aggregating over multiple independent external regret minimizers. In addition, the dynamics involve a fixed point operation—namely, the stationary distribution of a Markov chain—posing new challenges compared to the analysis of no-external-regret algorithms [Chen and Peng, 2020]. Finally, a very intriguing approach for obtaining near-optimal no-external-regret dynamics was recently introduced by Piliouras et al. [2021]. The main caveat of that result is that the dynamics they propose are *not* uncoupled, which has been a central desideratum in the line of work on no-regret learning in games. For this reason, the result in [Piliouras et al., 2021] is not directly comparable with the previous approaches.

## 2 Preliminaries

In this section we introduce the basic background on online optimization and learning in games. For a comprehensive treatment on the subject we refer the interested reader to the excellent book of Cesa-Bianchi and Lugosi [2006].

**Conventions** We denote by $\mathbb{N} = \{1, 2, \dots\}$ the set of natural numbers. We use the shorthand notation $[\![n]\!] \coloneqq \{1, 2, \dots, n\}$. Subscripts are typically used to indicate the player, or a parameter uniquely associated with a player (such as an action available to the player). On the other hand, superscripts are reserved almost exclusively for the (discrete) time index, which is represented via the variable $t$. Also, the $r$-th coordinate of a $d$-dimensional vector $\boldsymbol{x} \in \mathbb{R}^d$ is denoted by $\boldsymbol{x}[r]$. Finally, we let $\log(\cdot)$ be the natural logarithm.

### 2.1 Online Learning and Phi-Regret

Let $\mathcal{X} \subseteq \mathbb{R}^d$ be a nonempty convex and compact set of strategies, for some $d \in \mathbb{N}$. In the online learning framework the *learner* has to select at every iteration $t \in \mathbb{N}$ a *strategy* $\boldsymbol{x}^{(t)} \in \mathcal{X}$. Then, the environment—be it the "nature" or some "adversary"—returns a (linear) utility function $u^{(t)} : \mathcal{X} \ni \boldsymbol{x} \mapsto \langle \boldsymbol{x}, \boldsymbol{u}^{(t)} \rangle$, for some utility vector $\boldsymbol{u}^{(t)} \in \mathbb{R}^d$, so that the learner receives a utility of $\langle \boldsymbol{x}^{(t)}, \boldsymbol{u}^{(t)} \rangle$ at time $t$. In the *full information* model the learner receives as feedback the entire utility function, represented by $\boldsymbol{u}^{(t)}$. The canonical measure of performance in online learning is based on the notion of *regret*, or more generally, on *Phi-regret* [Greenwald and Jafari, 2003, Stoltz and Lugosi, 2007, Gordon et al., 2008]. Formally, for a set of transformations $\Phi : \mathcal{X} \to \mathcal{X}$, the $\Phi$-*regret* of a regret minimization algorithm $\mathfrak{R}$ up to a *time horizon* $T \in \mathbb{N}$ is defined as

$$\text{Reg}_\Phi^T \coloneqq \max_{\phi^* \in \Phi} \left\{ \sum_{t=1}^T \langle \phi^*(\boldsymbol{x}^{(t)}), \boldsymbol{u}^{(t)} \rangle \right\} - \sum_{t=1}^T \langle \boldsymbol{x}^{(t)}, \boldsymbol{u}^{(t)} \rangle. \tag{1}$$

Naturally, a broader collection of transformations leads to a stronger notion of hindsight rationality; canonical instantiations of Phi-regret include:

(i) *External regret* (denoted by $\text{Reg}$): $\Phi$ includes only *constant transformations*;

(ii) *Swap regret* (denoted by $\text{SwapReg}$): $\Phi$ includes *all* possible linear transformations.

As such, swap regret induces the more powerful notion of hindsight rationality. We point out that our main focus in this paper (Section 4) will be for the special case where $\mathcal{X}$ is the probability simplex. A crucial property of swap regret is that $\text{SwapReg} \geq 0$, as formalized below.

**Observation 2.1.** *Fix any time horizon $T \in \mathbb{N}$. For any sequence of utilities $\boldsymbol{u}^{(1)}, \dots, \boldsymbol{u}^{(T)}$ and any sequence of strategies $\boldsymbol{x}^{(1)}, \dots, \boldsymbol{x}^{(T)}$ it holds that $\text{SwapReg}^T \geq 0$.*

In proof, just consider the identity transformation $\Phi \ni \phi : \boldsymbol{x} \mapsto \boldsymbol{x}$ in (1). In contrast, this property does not necessarily hold for external regret.

Moreover, it will be convenient to model a regret minimization algorithm $\mathfrak{R}$ as a black box which interacts with its environment via the following two subroutines.

(i) $\mathfrak{R}.\text{NEXTSTRATEGY}()$: $\mathfrak{R}$ returns the next strategy of the learner;

(ii) $\mathfrak{R}.\text{OBSERVEUTILITY}(\boldsymbol{u})$: $\mathfrak{R}$ receives as feedback from the environment a utility vector $\boldsymbol{u}$, and may adapt its internal state accordingly.

## 2.2 No-Regret Learning and Correlated Equilibria

A fundamental connection ensures that as long as all players employ no-swap-regret learning dynamics (in the sense that $\text{SwapReg}^T = o(T)$), the average correlated distribution of play converges to the set of correlated equilibria [Hart and Mas-Colell, 2000, Foster and Vohra, 1997, Blum and Mansour, 2007]. Before we formalize this connection, let us first introduce some basic background on games.

**Finite Games**  Let $[\![n]\!] := \{1, 2, \ldots, n\}$ be the set of players, with $n \geq 2$. In a (finite) game, represented in *normal form*, each player $i \in [\![n]\!]$ has a finite set of actions $\mathcal{A}_i$; for notational simplicity, we will let $m_i := |\mathcal{A}_i| \geq 2$. For a given joint action profile $\boldsymbol{a} = (a_1, \ldots, a_n) \in \bigtimes_{i=1}^n \mathcal{A}_i$, the (normalized) utility received by player $i$ is given by some arbitrary function $u_i : \bigtimes_{i=1}^n \mathcal{A}_i \to [-1, 1]$. Players are allowed to randomize by selecting a (mixed) strategy $\boldsymbol{x}_i \in \Delta(\mathcal{A}_i) := \left\{ \boldsymbol{x} \in \mathbb{R}_{\geq 0}^{|\mathcal{A}_i|} : \sum_{a_i \in \mathcal{A}_i} \boldsymbol{x}[a_i] = 1 \right\}$; that is, a probability distribution over the available actions. For a joint strategy profile $\boldsymbol{x} = (\boldsymbol{x}_1, \ldots, \boldsymbol{x}_n)$, player $i$ receives an *expected utility* of $\mathbb{E}_{\boldsymbol{a} \sim \boldsymbol{x}}[u_i(\boldsymbol{a})] = \sum_{\boldsymbol{a} \in \mathcal{A}} u_i(\boldsymbol{a}) \prod_{j \in [\![n]\!]} \boldsymbol{x}_j[a_j]$.

In the problem of no-regret learning in games, every player receives as feedback at time $t \in \mathbb{N}$ a utility vector $\boldsymbol{u}_i^{(t)} \in \mathbb{R}^{|\mathcal{A}_i|}$, so that $\boldsymbol{u}_i^{(t)}[a_i] := u_i(a_i; \boldsymbol{x}_{-i}^{(t)}) := \mathbb{E}_{\boldsymbol{a}_{-i} \sim \boldsymbol{x}_{-i}}[u_i(a_i, \boldsymbol{a}_{-i})]$, for any $a_i \in \mathcal{A}_i$; here, we used the notation $\boldsymbol{a}_{-i}$ to represent the joint action profile excluding $i$'s component, and analogously for the notation $\boldsymbol{x}_{-i}$. *No other information is available to the player.* We are now ready to introduce the concept of a *correlated equilibrium* due to Aumann [1974].

**Definition 2.2** (Correlated Equilibrium [Aumann, 1974]). *A probability distribution $\boldsymbol{\mu}$ over $\bigtimes_{i=1}^n \mathcal{A}_i$ is an $\epsilon$-approximate correlated equilibrium, for $\epsilon \geq 0$, if for any player $i \in [\![n]\!]$ and any swap function $\phi_i : \mathcal{A}_i \to \mathcal{A}_i$,*

$$\mathbb{E}_{\boldsymbol{a} \sim \boldsymbol{\mu}}[u_i(\boldsymbol{a})] \geq \mathbb{E}_{\boldsymbol{a} \sim \boldsymbol{\mu}}[u_i(\phi_i(a_i), \boldsymbol{a}_{-i})] - \epsilon.$$

**Theorem 2.3** (Folklore). *Suppose that each player $i \in [\![n]\!]$ employs a no-swap-regret algorithm such that the cumulative swap regret up to time $T \in \mathbb{N}$ is upper bounded by $\text{SwapReg}_i^T$. Further, let $\boldsymbol{\mu}^{(t)} := \boldsymbol{x}_1^{(t)} \otimes \boldsymbol{x}_2^{(t)} \otimes \cdots \otimes \boldsymbol{x}_n^{(t)}$ be the product distribution at time $t \in [\![T]\!]$, and $\bar{\boldsymbol{\mu}} := \frac{1}{T} \sum_{t=1}^T \boldsymbol{\mu}^{(t)}$ be the average correlated distribution of play up to time $T$. Then, $\bar{\boldsymbol{\mu}}$ is a $\max_{i=1}^n \{\text{SwapReg}_i^T / T\}$-approximate correlated equilibrium.*

Consequently, a central challenge for correlated equilibria is that the rate of convergence is driven by the *maximum* of the swap regrets; this is in contrast to, for example, the rate of convergence of the (utilitarian) social welfare in *smooth games*, which is driven by the *sum* of the players' external regrets [Syrgkanis et al., 2015, Roughgarden, 2015].

## 3 Optimistic Learning with Self-Concordant Barriers

*Optimistic follow the regularizer leader (OFTRL)* [Syrgkanis et al., 2015] is a *predictive* variant of the standard FTRL paradigm. Specifically, OFTRL maintains an internal *prediction* vector $\boldsymbol{m}^{(t)} \in \mathbb{R}^d$, and can be expressed with the following update rule for $t \in \mathbb{N}$.

$$\boldsymbol{x}^{(t)} := \arg\max_{\boldsymbol{x} \in \mathcal{X}} \left\{ \Phi^{(t)}(\boldsymbol{x}) := \eta \left\langle \boldsymbol{x}, \boldsymbol{m}^{(t)} + \sum_{\tau=1}^{t-1} \boldsymbol{u}^{(\tau)} \right\rangle - \mathcal{R}(\boldsymbol{x}) \right\}; \qquad \text{(OFTRL)}$$

here, $\eta > 0$ serves as the *learning rate*, and $\mathcal{R}$ is the *regularizer*. For convenience, we also define $\boldsymbol{x}^{(0)} := \arg\min_{\boldsymbol{x} \in \mathcal{X}} \mathcal{R}(\boldsymbol{x})$. Unless specified otherwise, (OFTRL) will be instantiated with $\boldsymbol{m}^{(t)} := \boldsymbol{u}^{(t-1)}$, for $t \in \mathbb{N}$. (For convenience in the analysis, and without any loss, we assume that players initially obtain the utilities corresponding to the other players' strategies at time $t = 0$.)

In [Syrgkanis et al., 2015] the regularizer $\mathcal{R}$ was assumed to be 1-strongly convex with respect to some (static) norm $\| \cdot \|$ on $\mathbb{R}^d$. On the other hand, we are introducing an important twist: $\mathcal{R}$ will

be a *self-concordant barrier function* over $\mathcal{X}$.[3] In this context, we first extend (in Appendix B) the so-called RVU bound established in [Syrgkanis et al., 2015] under self-concordant regularization. More precisely, we assume that $\mathcal{X}$ has nonempty interior $\text{int}(\mathcal{X})$. Further, for $\boldsymbol{u} \in \mathbb{R}^d$ the primal local norm with respect to $\boldsymbol{x} \in \text{int}(\mathcal{X})$ is defined as $\|\boldsymbol{u}\|_{\boldsymbol{x}} := \sqrt{\boldsymbol{u}^\top \nabla^2 \mathcal{R}(\boldsymbol{x}) \boldsymbol{u}}$, while the dual norm is defined as $\|\boldsymbol{u}\|_{*,\boldsymbol{x}} := \sqrt{\boldsymbol{u}^\top (\nabla^2 \mathcal{R}(\boldsymbol{x}))^{-1} \boldsymbol{u}}$, assuming that $\mathcal{R}$ nondegenerate—in the sense that its Hessian is positive definite. Finally, for the purpose of the analysis, we let $\boldsymbol{g}^{(t)}$ denote the *be the leader* sequence (see (BTL) in Appendix B); no attempt was made to optimize universal constants.

**Theorem 3.1** (RVU for Self-Concordant Regularizers). *Suppose that $\mathcal{R}$ is a nondegenerate self-concordant function for $\text{int}(\mathcal{X})$. Moreover, let $\eta > 0$ be such that $\eta \|\boldsymbol{u}^{(t)} - \boldsymbol{m}^{(t)}\|_{*,\boldsymbol{x}^{(t)}} \leq \frac{1}{2}$ and $\eta \|\boldsymbol{m}^{(t)}\|_{*,\boldsymbol{g}^{(t-1)}} \leq \frac{1}{2}$ for all $t \in [\![T]\!]$. Then, the regret $\text{Reg}^T(\boldsymbol{x}^*)$ of (OFTRL) with respect to any comparator $\boldsymbol{x}^* \in \text{int}(\mathcal{X})$ under any sequence of utilities $\boldsymbol{u}^{(1)}, \ldots, \boldsymbol{u}^{(T)}$ can be bounded by*

$$\frac{\mathcal{R}(\boldsymbol{x}^*)}{\eta} + 2\eta \sum_{t=1}^T \|\boldsymbol{u}^{(t)} - \boldsymbol{m}^{(t)}\|_{*,\boldsymbol{x}^{(t)}}^2 - \frac{1}{4\eta} \sum_{t=1}^T \left( \|\boldsymbol{x}^{(t)} - \boldsymbol{g}^{(t)}\|_{\boldsymbol{x}^{(t)}}^2 + \|\boldsymbol{x}^{(t)} - \boldsymbol{g}^{(t-1)}\|_{\boldsymbol{g}^{(t-1)}}^2 \right).$$

Here, we also used the standard notation $\text{Reg}^T(\boldsymbol{x}^*) := \sum_{t=1}^T \langle \boldsymbol{x}^* - \boldsymbol{x}^{(t)}, \boldsymbol{u}^{(t)} \rangle$. Next, we instantiate Theorem 3.1 using the *log-barrier* on the (probability) simplex: $\mathcal{R}(\boldsymbol{x}) = -\sum_{r=1}^d \log(\boldsymbol{x}[r])$. While the probability simplex has empty interior, there is a simple transformation on the relative interior $\text{relint}(\Delta^d)$ that addresses that issue (see Appendix B).

**Corollary 3.2** (RVU for Log-Barrier on the Simplex). *Suppose that $\mathcal{R}$ is the log-barrier on the simplex and $\eta \leq \frac{1}{16}$. Then, the regret of (OFTRL) under any sequence of utilities $\boldsymbol{u}^{(1)}, \ldots, \boldsymbol{u}^{(T)}$ can be bounded as*

$$\text{Reg}^T(\boldsymbol{x}^*) \leq \frac{\mathcal{R}(\boldsymbol{x}^*)}{\eta} + 2\eta \sum_{t=1}^T \|\boldsymbol{u}^{(t)} - \boldsymbol{u}^{(t-1)}\|_{*,\boldsymbol{x}^{(t)}}^2 - \frac{1}{16\eta} \sum_{t=1}^T \|\boldsymbol{x}^{(t)} - \boldsymbol{x}^{(t-1)}\|_{\boldsymbol{x}^{(t-1)}}^2,$$

*for any $\boldsymbol{x}^* \in \text{relint}(\Delta^d)$, where $\|\boldsymbol{x}^{(t)} - \boldsymbol{x}^{(t-1)}\|_{\boldsymbol{x}^{(t-1)}}^2 := \sum_{r=1}^d \left( \frac{\boldsymbol{x}^{(t)}[r] - \boldsymbol{x}^{(t-1)}[r]}{\boldsymbol{x}^{(t-1)}[r]} \right)^2$.*

We remark that a similar regret bound for *optimistic mirror descent* [Rakhlin and Sridharan, 2013] under log-barrier regularization was shown by [Wei and Luo, 2018, Theorem 7].

## 4 Main Result

In this section we sketch the proof of our main result, namely Theorem 1.1, leading to Corollaries 1.2 and 1.3; detailed proofs are deferred to Appendix C. In this context, we first employ the general template of Blum and Mansour [2007] for constructing a no-swap-regret minimizer $\mathfrak{R}_{swap}$ over the simplex. We proceed with a brief overview of their construction (summarized in Algorithm 1). In the sequel, we first perform the analysis from the perspective of a single player, without explicitly indicating so in our notation.

**The Algorithm of Blum and Mansour** Blum and Mansour [2007] construct a "master" regret minimization algorithm $\mathfrak{R}_{swap}$ by maintaining a separate and independent external regret minimizer $\mathfrak{R}_a$ for every action $a \in \mathcal{A}$. To compute the next strategy, $\mathfrak{R}_{swap}$ first obtains the strategy $\boldsymbol{x}_a^{(t)} \in \Delta(\mathcal{A})$ of $\mathfrak{R}_a$, for every $a \in \mathcal{A}$. Then, a (row) stochastic matrix $\mathbf{Q}^{(t)} \in \mathbb{S}^{|\mathcal{A}|}$ is constructed, so that the row associated with action $a \in \mathcal{A}$ is equal to the distribution $\boldsymbol{x}_a^{(t)}$, while $\mathfrak{R}_{swap}$ outputs as the next strategy $\boldsymbol{x}^{(t)} \in \Delta(\mathcal{A})$ any stationary distribution of $\mathbf{Q}^{(t)}$; that is, $(\mathbf{Q}^{(t)})^\top \boldsymbol{x}^{(t)} = \boldsymbol{x}^{(t)}$. Next, upon observing a utility $\boldsymbol{u}^{(t)} \in \mathbb{R}^{|\mathcal{A}|}$, $\mathfrak{R}_{swap}$ forwards to each individual regret minimizer $\mathfrak{R}_a$ the utility $\boldsymbol{u}_a^{(t)} := \boldsymbol{u}^{(t)} \boldsymbol{x}^{(t)}[a] \in \mathbb{R}^{|\mathcal{A}|}$. This construction is summarized in Algorithm 1.

---

[3]To keep the exposition reasonably self-contained, we give an overview of self-concordant barriers in Appendix A.

---

**Algorithm 1:** Blum and Mansour [2007]

---

**Input:** A set of external regret minimizers $\{\mathfrak{R}_a\}_{a \in \mathcal{A}}$, each for the simplex $\Delta(\mathcal{A})$

---

1 **function** NEXTSTRATEGY()
2     $\mathbf{Q}^{(t)} \leftarrow \mathbf{0} \in \mathbb{R}^{|\mathcal{A}| \times |\mathcal{A}|}$
3     **for** $a \in \mathcal{A}$ **do**
4         $\mathbf{Q}^{(t)}[a, \cdot] \leftarrow \mathfrak{R}_a.\text{NEXTSTRATEGY}()$
5     $\boldsymbol{x}^{(t)} \leftarrow \text{STATIONARYDISTRIBUTION}(\mathbf{Q}^{(t)})$
6     **return** $\boldsymbol{x}^{(t)}$

---

7 **function** OBSERVEUTILITY($\boldsymbol{u}^{(t)}$)
8     **for** $a \in \mathcal{A}$ **do**
9         $\mathfrak{R}_a.\text{OBSERVEUTILITY}(\boldsymbol{x}^{(t)}[a]\,\boldsymbol{u}^{(t)})$

---

Blum and Mansour [2007] showed that this algorithm guarantees *no-swap-regret* as long as each individual regret minimizer has sublinear *external regret*; this is formalized in the theorem below.

**Theorem 4.1** (From External to Swap Regret [Blum and Mansour, 2007]). *Let* $\text{SwapReg}^T$ *be the swap regret of* $\mathfrak{R}_{swap}$ *and* $\text{Reg}_a^T$ *be the external regret of* $\mathfrak{R}_a$, *for each* $a \in \mathcal{A}$, *up to time* $T \in \mathbb{N}$. *Then,*

$$\text{SwapReg}^T = \sum_{a \in \mathcal{A}} \text{Reg}_a^T .$$

In this context, we will instantiate each individual regret minimizer $\mathfrak{R}_a$ with (OFTRL) under log-barrier regularization—and the same learning rate $\eta > 0$. We will refer to the resulting algorithm as `BM-OFTRL-LogBar`. A central ingredient in our proof of Theorem 1.1 is to establish that the resulting no-swap-regret algorithm $\mathfrak{R}_{swap}$ will enjoy an RVU bound, as stated in Theorem 4.3. To this end, we first apply Corollary 3.2 for each individual regret minimizer $\mathfrak{R}_a$, implying that $\text{SwapReg}^T = \sum_{a \in \mathcal{A}} \text{Reg}_a^T$ (by Theorem 4.1) is upper bounded as

$$\text{SwapReg}^T \leq \frac{2m^2 \log T}{\eta} + 2\eta \sum_{a \in \mathcal{A}} \sum_{t=1}^{T} \|\boldsymbol{u}^{(t)} \boldsymbol{x}^{(t)}[a] - \boldsymbol{u}^{(t-1)} \boldsymbol{x}^{(t-1)}[a]\|_{*,\boldsymbol{x}_a^{(t)}}^2$$

$$- \frac{1}{16\eta} \sum_{a \in \mathcal{A}} \sum_{t=1}^{T} \|\boldsymbol{x}_a^{(t)} - \boldsymbol{x}_a^{(t-1)}\|_{\boldsymbol{x}_a^{(t-1)}}^2. \tag{2}$$

The $\log T$ factor derives from the diameter of the log-barrier regularizer (see Theorem A.9), and appears to be unavoidable using our approach. Now the crux in establishing an RVU bound for $\mathfrak{R}_{swap}$ is to upper bound the last term in (2) in terms of the "movement" of the stationary distribution. This is exactly where the local norm induced by the log-barrier turns out to be crucial, leading to the following key technical ingredient.

**Lemma 4.2.** *Suppose that each regret minimizer* $\mathfrak{R}_a$ *employs* (OFTRL) *with log-barrier regularization and* $\eta \leq \frac{1}{16}$. *Then, for any* $t \in \mathbb{N}$,

$$\|\boldsymbol{x}^{(t)} - \boldsymbol{x}^{(t-1)}\|_1^2 \leq 64|\mathcal{A}| \sum_{a \in \mathcal{A}} \|\boldsymbol{x}_a^{(t)} - \boldsymbol{x}_a^{(t-1)}\|_{\boldsymbol{x}_a^{(t-1)}}^2.$$

Intuitively, this lemma ensures that the "movement" of the stationary distribution is smooth in terms of the "movement" of each row of the transition matrix $\mathbf{Q}^{(t)}$. To show this, we use the Markov chain tree theorem (Theorem C.3), which provides a closed-form combinatorial formula for the stationary distribution of an ergodic Markov chain, along with the fact that the log-barrier regularizer guarantees "multiplicative stability" of the iterates (Corollary C.1). While similar in spirit results have been documented in the literature for dynamics akin to MWU [Candogan et al., 2013, Chen and Peng, 2020], our proof of Lemma 4.2 crucially hinges on the local norm induced by the log-barrier regularizer. Thus, we are now ready to derive an RVU bound for swap regret.

**Theorem 4.3** (RVU Bound for Swap Regret). *Suppose that each* $\mathfrak{R}_a$ *employs* (OFTRL) *with log-barrier regularization and* $\eta \leq \frac{1}{128\sqrt{m}}$. *Then, for* $T \geq 2$, *the swap regret of* $\mathfrak{R}_{swap}$ *is bounded*

*as*

$$\mathrm{SwapReg}^T \leq \frac{2m^2 \log T}{\eta} + 4\eta \sum_{t=1}^{T} \|\boldsymbol{u}^{(t)} - \boldsymbol{u}^{(t-1)}\|_\infty^2 - \frac{1}{2048m\eta} \sum_{t=1}^{T} \|\boldsymbol{x}^{(t)} - \boldsymbol{x}^{(t-1)}\|_1^2.$$

This theorem follows directly from (2) and Lemma 4.2. So far we have focused on bounding the swap regret of each player when faced against arbitrary utilities. Next, we use Theorem 4.3 to establish a new fundamental property when all players employ the dynamics. Our proof crucially relies on the seemingly insignificant fact that $\mathrm{SwapReg}_i^T \geq 0$ (recall Observation 2.1).

**Theorem 4.4** (Log-Bounded Second-Order Path Lengths). *Suppose that each player $i \in [\![n]\!]$ employs* BM-OFTRL-LogBar *with* $\eta = \frac{1}{128(n-1)\max_{j \in [\![n]\!]}\{\sqrt{m_j}\}}$. *Then, for $T \geq 2$,*

$$\sum_{i=1}^{n} \sum_{t=1}^{T} \|\boldsymbol{x}_i^{(t)} - \boldsymbol{x}_i^{(t-1)}\|_1^2 \leq 8192 \max_{i \in [\![n]\!]}\{m_i\} \sum_{i=1}^{n} m_i^2 \log T.$$

*Proof.* Consider any player $i \in [\![n]\!]$. Given that $|u_i(\boldsymbol{a})| \leq 1$, for any $\boldsymbol{a} \in \mathcal{A}$ (by the normalization assumption), we have that for any $t \in [\![T]\!]$,

$$\|\boldsymbol{u}_i^{(t)} - \boldsymbol{u}_i^{(t-1)}\|_\infty \leq \sum_{\boldsymbol{a}_{-i} \in \mathcal{A}_{-i}} \left| \prod_{j \neq i} \boldsymbol{x}_j^{(t)}[a_j] - \prod_{j \neq i} \boldsymbol{x}_j^{(t-1)}[a_j] \right| \leq \sum_{j \neq i} \|\boldsymbol{x}_j^{(t)} - \boldsymbol{x}_j^{(t-1)}\|_1,$$

where we used that the total variation distance between two product distributions is bounded by the sum of the total variations of each individual marginal distribution [Hoeffding and Wolfowitz, 1958]. Thus,

$$\left( \|\boldsymbol{u}_i^{(t)} - \boldsymbol{u}_i^{(t-1)}\|_\infty \right)^2 \leq \left( \sum_{j \neq i} \|\boldsymbol{x}_j^{(t)} - \boldsymbol{x}_j^{(t-1)}\|_1 \right)^2 \leq (n-1) \sum_{j \neq i} \|\boldsymbol{x}_j^{(t)} - \boldsymbol{x}_j^{(t-1)}\|_1^2.$$

As a result, using Theorem 4.3 we conclude that $\sum_{i=1}^{n} \mathrm{SwapReg}_i^T$ can be upper bounded by

$$2 \log T \sum_{i=1}^{n} \frac{m_i^2}{\eta} + 4\eta(n-1) \sum_{i=1}^{n} \sum_{j \neq i} \sum_{t=1}^{T} \|\boldsymbol{x}_j^{(t)} - \boldsymbol{x}_j^{(t-1)}\|_1^2 - \sum_{i=1}^{n} \frac{1}{2048m_i\eta} \sum_{t=1}^{T} \|\boldsymbol{x}_i^{(t)} - \boldsymbol{x}_i^{(t-1)}\|_1^2$$

$$= 2 \log T \sum_{i=1}^{n} \frac{m_i^2}{\eta} + \sum_{i=1}^{n} \left( 4\eta(n-1)^2 - \frac{1}{2048m_i\eta} \right) \sum_{t=1}^{T} \|\boldsymbol{x}_i^{(t)} - \boldsymbol{x}_i^{(t-1)}\|_1^2$$

$$\leq 2 \log T \sum_{i=1}^{n} \frac{m_i^2}{\eta} - \frac{1}{4096} \sum_{i=1}^{n} \frac{1}{m_i\eta} \sum_{t=1}^{T} \|\boldsymbol{x}_i^{(t)} - \boldsymbol{x}_i^{(t-1)}\|_1^2,$$

since $\eta = \frac{1}{128(n-1)\max_{j \in [\![n]\!]}\{\sqrt{m_j}\}}$. But, given that $0 \leq \sum_{i=1}^{n} \mathrm{SwapReg}_i^T$, we conclude that

$$\frac{1}{\max_{i \in [\![n]\!]}\{\sqrt{m_i}\}} \sum_{i=1}^{n} \sum_{t=1}^{T} \|\boldsymbol{x}_i^{(t)} - \boldsymbol{x}_i^{(t-1)}\|_1^2 \leq 8192 \max_{i \in [\![n]\!]}\{\sqrt{m_i}\} \sum_{i=1}^{n} m_i^2 \log T.$$

$\square$

We are not aware of even $o(T)$ bounds for the second-order path lengths in prior works (using a time-invariant learning rate), except in very restricted classes of games such as zero-sum and potential games [Anagnostides et al., 2022c]. An example of the implication of Theorem 4.4 in a variant of *Shapley's game* [Shapley, 1964, Daskalakis et al., 2010] is illustrated in Figure 1. Although the dynamics appear to cycle, and the Nash gap—the maximum of the best response gaps—is always large, the players are changing their (mixed) strategies with gradually diminishing speed; further discussion and experiments are included in Appendix D.

As an immediate consequence, combining Theorem 4.4 with Theorem 4.3 implies near-optimal individual swap regret.

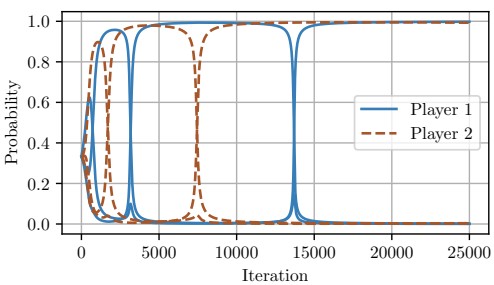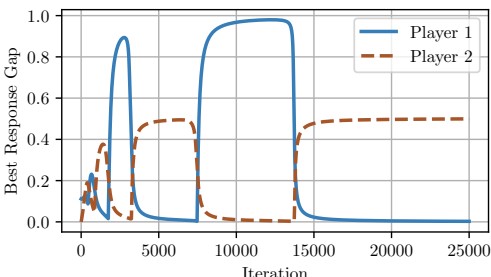

Figure 1: The trajectories of the `BM-OFTRL-LogBar` algorithm.

**Corollary 4.5** (Near-Optimal Individual Swap Regret). *Suppose that all players use* `BM-OFTRL-LogBar` *with* $\eta = \frac{1}{128(n-1)\max_{i \in [\![n]\!]}\{\sqrt{m_i}\}}$*. Then, the individual swap regret* $\mathrm{SwapReg}_i^T$ *up to time* $T \geq 2$ *of each player* $i \in [\![n]\!]$ *can be bounded as*

$$\mathrm{SwapReg}_i^T \leq 256 \max_{j \in [\![n]\!]}\{\sqrt{m_j}\} \left( (n-1)m_i^2 + \sum_{j=1}^{n} m_j^2 \right) \log T.$$

We point out that our distributed protocol makes the very mild assumption that each player knows an upper bound on the total number of players and the maximum number of actions in order to appropriately tune the learning rate. Further, as is the case with the result in [Daskalakis et al., 2021], the individual regret of each player predicted by Corollary 4.5 grows linearly with the number of players. This can be unsatisfactory in games with a large number of players—*i.e.*, $n \gg 1$. For this reason, in Theorem C.4 we refine and improve the guarantee of Corollary 4.5 in games where the utility of each player depends only on a small number of other players, and each player's actions only affect a small number of others players; no other constraint is imposed on the game. Understanding whether the linear dependence on $n$ is necessary to obtain near-optimal (swap) regret is left as an interesting question for future work.

Finally, we adapt the learning dynamics so that each player enjoys at the same time near-optimal swap regret in the adversarial regime as well.

**Corollary 4.6** (Adversarial Robustness). *There exist dynamics such that when all players follow them the individual swap regret of each player grows as in Corollary 4.5. Moreover, when faced against adversarial utilities, such that* $\|\boldsymbol{u}_i^{(t)}\|_\infty \leq 1$ *for all* $t \in [\![T]\!]$*, the algorithm guarantees that*

$$\mathrm{SwapReg}_i^T \leq 256 \max_{j \in [\![n]\!]}\{\sqrt{m_j}\} \left( (n-1)m_i^2 + \sum_{j=1}^{n} m_j^2 \right) \log T + 2\sqrt{m_i \log m_i T} + 2.$$

Our adaptation is particularly natural: If all players follow the prescribed protocol, Theorem 4.4 implies that the observed utilities of each player $i$ will be such that $\sum_{\tau=1}^{t} \|\boldsymbol{u}_i^{(\tau)} - \boldsymbol{u}_i^{(\tau-1)}\|_\infty = O(\log t)$. So, if at any time the player identifies that the previous condition was violated, it suffices to switch to a no-swap-regret minimizer (such as BM-MWU) tuned to face adversarial losses—in which case it is crucial to use a vanishing learning rate $\eta = O(1/\sqrt{T})$.

*Remark* 4.7 (Numerical Precision). As is standard, we assumed that the iterates of (OFTRL) were computed exactly, without taking into account issues relating to numerical precision. To justify this, one can use *Damped Newton's method* in order to determine an $\epsilon$-nearby point to the optimal in $O(\log\log(1/\epsilon))$ iterations [Nemirovski and Todd, 2008]. This would extend all the regret bounds with up to an $O(\epsilon T)$ error. So, with only $O(\log\log T)$ repetitions of Damped Newton's method (per iteration) the error in the regret bounds becomes $O(1)$, and all of our guarantees immediately extend; see [Farina et al., 2022, Appendix A.5] for an analogous extension under approximate iterates.

## 5    Discussion

Our main contribution in this paper was to establish a fundamental new property characterizing the trajectories of certain uncoupled no-regret learning dynamics, summarized in Theorem 1.1. This property directly guarantees the best known and near-optimal bound of $O(\log T)$ for the swap regret incurred by each player in a general multiplayer game. Investigating further consequences of Theorem 1.1 is an interesting direction for the future. We also believe that our framework could have new implications for learning in games with partial information; *e.g.*, see [Wei and Luo, 2018]. Another interesting avenue is to extend our scope to more general and combinatorial sets beyond the probability simplex, in order to (efficiently) encompass, for example, games in *extensive form*.

Further, our no-swap-regret learning dynamics have external regret trivially bounded according to Corollary 4.5. Consequently, our construction yields no-external-regret learning dynamics with a more favorable dependence on $T$ compared to [Daskalakis et al., 2021] ($\log T$ compared to the $\log^4(T)$ of the latter), but with a worse dependence on the number of actions (polynomial rather than logarithmic). Our method also has higher per-iteration complexity. For these reasons, extending the scope of our framework beyond self-concordant regularization is an important direction for future research. Indeed, we conjecture that OMWU has *bounded* second-order path lengths, a property that would imply the first uncoupled learning dynamics with bounded regret, but establishing that likely requires new insights.

## Acknowledgments

We are thankful to the anonymous NeurIPS reviewers for many helpful comments. Ioannis Anagnostides is grateful to Dimitris Achlioptas for helpful discussions. Christian Kroer is supported by the Office of Naval Research Young Investigator Program under grant N00014-22-1-2530. Haipeng Luo is supported by the National Science Foundation under grant IIS-1943607 (part of this work was done while he was visiting the Simons Institute for the Theory of Computing). Tuomas Sandholm is supported by the National Science Foundation under grants IIS-1901403 and CCF-1733556.

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
