# A  Preliminaries on Self-Concordant Barriers

In this section we provide the necessary background on self-concordant barriers. For a more comprehensive overview on the theory of self-concordant barriers and their role in interior-point methods we refer to the book of Nesterov [2004], the lecture notes of Nemirovski [2004], as well as the survey of Nemirovski and Todd [2008]. We start this section by introducing the central concept of a *self-concordant function*.

## A.1  Self-Concordant Functions

**Definition A.1** (Self-Concordant Function). Let $Q \subseteq \mathbb{R}^d$ be a nonempty open and convex set. A convex function $f : Q \to \mathbb{R}$ in $\mathcal{C}^3$ is called *self-concordant* on $Q$ if it satisfies the following properties.

   (i) (Barrier property) For every sequence $(\boldsymbol{x}_i \in Q)_{i=1}^{\infty}$ converging to a boundary point of $Q$ as $i \to \infty$ it holds that $f(\boldsymbol{x}_i) \to \infty$;

   (ii) (Differential inequality of self-concordance) $f$ satisfies the inequality

$$|D^3 f(\boldsymbol{x})[\boldsymbol{u}, \boldsymbol{u}, \boldsymbol{u}]| \le 2 \left( D^2 f(\boldsymbol{x})[\boldsymbol{u}, \boldsymbol{u}] \right)^{3/2}, \tag{3}$$

   for all $\boldsymbol{x} \in Q$ and $\boldsymbol{u} \in \mathbb{R}^d$.

In (3) we used the notation

$$D^k f(\boldsymbol{x})[\boldsymbol{u}_1, \ldots, \boldsymbol{u}_k] := \left. \frac{\partial^k}{\partial s_1 \ldots \partial s_k} \right|_{s_1 = \cdots = s_k = 0} f(\boldsymbol{x} + s_1 \boldsymbol{u}_1 + \cdots + s_k \boldsymbol{u}_k)$$

to denote the $k$-th-order differential of $f$ at point $\boldsymbol{x}$ along the directions $\boldsymbol{u}_1, \boldsymbol{u}_2, \ldots, \boldsymbol{u}_k$. Self-concordance, in the sense of Definition A.1, basically imposes a Lipschitz-continuity condition on the Hessian of $f$, but with respect to the *local norm* induced by the Hessian itself [Nemirovski, 2004]. One may allow (3) to hold with a multiplicative factor $M_f \ge 0$ on the right hand side, in which case $f$ is said to be self-concordant with parameter $M_f$; unless explicitly specified otherwise, it will be assumed that $M_f = 1$. As a concrete example, we point out that the logarithmic barrier for the nonnegative ray, namely the univariate function $f : (0, +\infty) \ni x \mapsto -\log x$, is self-concordant (with parameter $M_f = 1$).

A crucial fact is that self-concordance is preserved under any linear perturbation, as can be verified directly from Definition A.1. We also point out a certain property which will be useful when composing different functions, and is also an immediate consequence of Definition A.1.

**Lemma A.2** ([Nemirovski, 2004]). *Let $f_i$ be self-concordant on $\operatorname{dom} f_i$, for all $i \in [\![k]\!]$. Then, assuming that $\operatorname{dom} f := \cap_{i=1}^{k} \operatorname{dom} f_i \ne \emptyset$, the function $f(\boldsymbol{x}) := \sum_{i=1}^{k} f_i(\boldsymbol{x})$ is self-concordant.*

## A.2  Useful Inequalities

Let $f$ be a self-concordant function. In the sequel we will tacitly assume that $f$ is *nondegenerate*, in the sense that the Hessian $\nabla^2 f(\boldsymbol{x})$ is positive definite, for any $\boldsymbol{x} \in \operatorname{dom} f$. In this context, we define $\|\boldsymbol{u}\|_{f,\boldsymbol{x}} := \sqrt{\boldsymbol{u}^\top \nabla^2 f(\boldsymbol{x}) \boldsymbol{u}}$ to be the *(primal) local norm* of direction $u$ induced by $f$ at point $\boldsymbol{x} \in \operatorname{dom} f$. (It is easy to verify that $\|\boldsymbol{u}\|_{f,\boldsymbol{x}}$ indeed satisfies the axioms of a norm.) To lighten our notation, we will oftentimes simply write $\|\boldsymbol{u}\|_{\boldsymbol{x}}$ when the underlying self-concordant function is clear from the context. The following inequality will be used to derive quadratic growth bounds with respect to the minimum of a self-concordant function.

**Lemma A.3** ([Nesterov, 2004]). *Let $f$ be a self-concordant function. Then, for any $\boldsymbol{x}, \widetilde{\boldsymbol{x}} \in \operatorname{dom} f$,*

$$f(\widetilde{\boldsymbol{x}}) \ge f(\boldsymbol{x}) + \langle \nabla f(\boldsymbol{x}), \widetilde{\boldsymbol{x}} - \boldsymbol{x} \rangle + \omega \left( \|\widetilde{\boldsymbol{x}} - \boldsymbol{x}\|_{\boldsymbol{x}} \right),$$

*where $\omega(s) := s - \log(1 + s)$.*

It will be convenient to use a quadratic lower bound for $\omega(s)$, as implied by the following simple fact.

**Fact A.4.** *Let $\omega(s) = s - \log(1 + s)$. Then,*

$$\omega(s) \ge \frac{s^2}{2(1 + s)}.$$

*In particular, for $s \in [0, 1]$ it holds that $\omega(s) \ge \frac{s^2}{4}$.*

Next, let us consider the optimization problem associated with the minimization of a self-concordant function, namely

$$\min\{f(\boldsymbol{x}) : \boldsymbol{x} \in \operatorname{dom} f\}, \tag{4}$$

for a self-concordant $f$. The *Newton Decrement* of $f$ at point $\boldsymbol{x} \in \operatorname{dom} f$ is defined as

$$\lambda(\boldsymbol{x}, f) := \|\nabla f(\boldsymbol{x})\|_{*,\boldsymbol{x}} = \sqrt{(\nabla f(\boldsymbol{x}))^\top (\nabla^2 f(\boldsymbol{x}))^{-1} \nabla f(\boldsymbol{x})}.$$

The following result guarantees (existence and) uniqueness for the optimization problem (4).

**Lemma A.5** ([Nesterov, 2004]). *Let $f$ be a self-concordant function such that $\lambda(\boldsymbol{x}, f) < 1$, for some $\boldsymbol{x} \in \operatorname{dom} f$. Then, the optimization problem* (4) *has a unique solution.*

Assuming that $\mathcal{X}$ is a convex and compact set with nonempty interior, we will also use the following important fact.

**Lemma A.6** ([Nemirovski, 2004]). *Let $f : \operatorname{int}(\mathcal{X}) \to \mathbb{R}$ be a self-concordant function with $\boldsymbol{x}^* := \arg\min_{\boldsymbol{x}} f(\boldsymbol{x})$, and some $\boldsymbol{x} \in \operatorname{int}(\mathcal{X})$. Then, if $\lambda(\boldsymbol{x}, f) \leq \frac{1}{2}$,*

$$\|\boldsymbol{x} - \boldsymbol{x}^*\|_{\boldsymbol{x}} \leq 2\lambda(\boldsymbol{x}, f);$$
$$\|\boldsymbol{x} - \boldsymbol{x}^*\|_{\boldsymbol{x}^*} \leq 2\lambda(\boldsymbol{x}, f).$$

### A.3 Self-Concordant Barriers

Next, we introduce the concept of a *self-concordant barrier*.

**Definition A.7** (Self-Concordant Barrier). Let $\mathcal{X} \subseteq \mathbb{R}^d$ be a convex and compact set with nonempty interior $\operatorname{int}(\mathcal{X})$ (domain). A function $f : \operatorname{int}(\mathcal{X}) \to \mathbb{R}$ is called a $\theta$-*self-concordant barrier* for $\mathcal{X}$ if

(i) $f$ is self-concordant on $\operatorname{int}(\mathcal{X})$; and

(ii) for all $\boldsymbol{x} \in \operatorname{int}(\mathcal{X})$ and $\boldsymbol{u} \in \mathbb{R}^d$,

$$|Df(\boldsymbol{x})[\boldsymbol{u}]| \leq \theta^{1/2} \left(D^2 f(\boldsymbol{x})[\boldsymbol{u}, \boldsymbol{u}]\right)^{1/2}. \tag{5}$$

We note that (5) imposes that $f$ is Lipshitz continuous with parameter $\theta^{1/2}$, but with respect to the local Euclidean metric induced by the Hessian. As an example, it is immediate to see that the function $\mathcal{R}(x) := -\log x$ is a 1-self-concordant barrier for the nonnegative ray. The following lemma will be useful when composing self-concordant barriers.

**Lemma A.8** ([Nesterov, 2004]). *Let $f_i$ be a $\theta_i$-self-concordant barrier for the compact and convex domain $\mathcal{X}_i \subseteq \mathbb{R}^d$, for all $i \in [\![k]\!]$. If the set $\mathcal{X} := \cap_{i \in [\![k]\!]} \mathcal{X}_i$ has nonempty interior, the function $f(\boldsymbol{x}) := \sum_{i=1}^k f_i(\boldsymbol{x})$ is a $\left(\sum_{i=1}^k \theta_i\right)$-self-concordant barrier for $\mathcal{X}$.*

**Minkowski Function**   Finally, we will require the fact that a self-concordant barrier does not grow overly quickly close to the boundary of $\mathcal{X}$. In particular, the growth is only *logarithmic* as a function of the inverse distance from the boundary. To formalize this, let us introduce the *Minkowski function* on $\mathcal{X}$, defined as follows.

$$\pi(\widetilde{\boldsymbol{x}}; \boldsymbol{x}) = \inf \left\{ s \geq 0 : \boldsymbol{x} + s^{-1}(\widetilde{\boldsymbol{x}} - \boldsymbol{x}) \in \mathcal{X} \right\}.$$

We remark that $\pi(\widetilde{\boldsymbol{x}}; \boldsymbol{x}) \in [0, 1]$. When $\boldsymbol{x}$ is the "center" of $\mathcal{X}$, $\pi(\widetilde{\boldsymbol{x}}; \boldsymbol{x})$ can be thought of as the distance of $\widetilde{\boldsymbol{x}}$ from the boundary of $\mathcal{X}$. In this context, we will use the following theorem.

**Theorem A.9.** *For any $\theta$-self-concordant barrier $\mathcal{R}$ on $\mathcal{X}$ and $\boldsymbol{x}, \widetilde{\boldsymbol{x}} \in \operatorname{int}(\mathcal{X})$,*

$$\mathcal{R}(\widetilde{\boldsymbol{x}}) - \mathcal{R}(\boldsymbol{x}) \leq \theta \log \left( \frac{1}{1 - \pi(\widetilde{\boldsymbol{x}}; \boldsymbol{x})} \right).$$

# B   RVU Bounds under Self-Concordant Barriers

In this section we establish the RVU property [Syrgkanis et al., 2015] for (OFTRL) when the regularizer is a self-concordant function. The main result of this section is Theorem 3.1, while Corollary 3.2 is an instantiation on the probability simplex.

As usual, for the purpose of our analysis we consider the auxiliary *be the leader (BTL) sequence*, defined for $t \in \mathbb{N} \cup \{0\}$ as follows.

$$\boldsymbol{g}^{(t)} := \arg\max_{\boldsymbol{g} \in \mathcal{X}} \left\{ \Psi^{(t)}(\boldsymbol{g}) := \eta \left\langle \boldsymbol{g}, \sum_{\tau=1}^{t} \boldsymbol{u}^{(\tau)} \right\rangle - \mathcal{R}(\boldsymbol{g}) \right\}. \tag{BTL}$$

By convention, we have let $\boldsymbol{g}^{(0)} := \arg\min_{\boldsymbol{g} \in \mathcal{X}} \mathcal{R}(\boldsymbol{g})$. We also remark that, as long as $\eta \| \boldsymbol{u}^{(t)} - \boldsymbol{m}^{(t)} \|_{*,\boldsymbol{x}^{(t)}} \leq \frac{1}{2}$ and $\eta \| \boldsymbol{m}^{(t)} \|_{*,\boldsymbol{g}^{(t-1)}} \leq \frac{1}{2}$, for all $t \in [\![T]\!]$, both (BTL) and (OFTRL) are well-posed, as can be verified using Lemma A.5 (see Lemma B.2). For convenience, and without any loss of generality, in the sequel it is assumed that $\mathcal{R}$ is normalized so that $\min_{\boldsymbol{x}} \mathcal{R}(\boldsymbol{x}) = 0$. We are now ready to establish the following theorem.

**Theorem B.1.** *Suppose that $\mathcal{R}$ is a nondegenerate self-concordant function for $\mathrm{int}(\mathcal{X})$, and let $\eta > 0$ be such that $\eta \| \boldsymbol{u}^{(t)} - \boldsymbol{m}^{(t)} \|_{*,\boldsymbol{x}^{(t)}} \leq \frac{1}{2}$ and $\eta \| \boldsymbol{m}^{(t)} \|_{*,\boldsymbol{g}^{(t-1)}} \leq \frac{1}{2}$, for all $t \in [\![T]\!]$. Then, the regret of (OFTRL) with respect to any $\boldsymbol{x}^* \in \mathrm{int}(\mathcal{X})$ and under any sequence of utilities $\boldsymbol{u}^{(1)}, \dots, \boldsymbol{u}^{(T)}$ can be bounded as*

$$\mathrm{Reg}^T(\boldsymbol{x}^*) \leq \frac{\mathcal{R}(\boldsymbol{x}^*)}{\eta} + \sum_{t=1}^{T} \| \boldsymbol{u}^{(t)} - \boldsymbol{m}^{(t)} \|_{*,\boldsymbol{x}^{(t)}} \| \boldsymbol{x}^{(t)} - \boldsymbol{g}^{(t)} \|_{\boldsymbol{x}^{(t)}}$$

$$- \frac{1}{\eta} \sum_{t=1}^{T} \left( \omega(\| \boldsymbol{x}^{(t)} - \boldsymbol{g}^{(t)} \|_{\boldsymbol{x}^{(t)}}) + \omega(\| \boldsymbol{x}^{(t)} - \boldsymbol{g}^{(t-1)} \|_{\boldsymbol{g}^{(t-1)}}) \right),$$

*where $\omega(\cdot)$ is defined as in Lemma A.3.*

*Proof.* The proof proceeds similarly to [Syrgkanis et al., 2015, Theorem 19]. The first observation is that

$$\langle \boldsymbol{x}^* - \boldsymbol{x}^{(t)}, \boldsymbol{u}^{(t)} \rangle = \langle \boldsymbol{g}^{(t)} - \boldsymbol{x}^{(t)}, \boldsymbol{u}^{(t)} - \boldsymbol{m}^{(t)} \rangle + \langle \boldsymbol{g}^{(t)} - \boldsymbol{x}^{(t)}, \boldsymbol{m}^{(t)} \rangle + \langle \boldsymbol{x}^* - \boldsymbol{g}^{(t)}, \boldsymbol{u}^{(t)} \rangle.$$

Given that $\langle \boldsymbol{g}^{(t)} - \boldsymbol{x}^{(t)}, \boldsymbol{u}^{(t)} - \boldsymbol{m}^{(t)} \rangle \leq \| \boldsymbol{u}^{(t)} - \boldsymbol{m}^{(t)} \|_{*,\boldsymbol{x}^{(t)}} \| \boldsymbol{x}^{(t)} - \boldsymbol{g}^{(t)} \|_{\boldsymbol{x}^{(t)}}$, by Hölder's inequality, it suffices to prove that for any $T \in \mathbb{N}$ and $\boldsymbol{x}^* \in \mathrm{int}(\mathcal{X})$,

$$\sum_{t=1}^{T} (\langle \boldsymbol{g}^{(t)} - \boldsymbol{x}^{(t)}, \boldsymbol{m}^{(t)} \rangle + \langle \boldsymbol{x}^* - \boldsymbol{g}^{(t)}, \boldsymbol{u}^{(t)} \rangle) \leq \frac{\mathcal{R}(\boldsymbol{x}^*)}{\eta}$$

$$- \frac{1}{\eta} \sum_{t=1}^{T} \left( \omega(\| \boldsymbol{x}^{(t)} - \boldsymbol{g}^{(t)} \|_{\boldsymbol{x}^{(t)}}) + \omega(\| \boldsymbol{x}^{(t)} - \boldsymbol{g}^{(t-1)} \|_{\boldsymbol{g}^{(t-1)}}) \right). \tag{6}$$

We will establish this claim via induction. For convenience, we use as base for the induction the case where $T = 0$, in which case (6) holds trivially since $\mathcal{R}(\boldsymbol{x}^*) \geq 0$ for any $\boldsymbol{x}^* \in \mathrm{int}(\mathcal{X})$.[4] Now for the inductive step, assume that for some $T \in \{0, 1, \dots\}$,

$$\sum_{t=1}^{T} (\langle \boldsymbol{g}^{(t)} - \boldsymbol{x}^{(t)}, \boldsymbol{m}^{(t)} \rangle - \langle \boldsymbol{g}^{(t)}, \boldsymbol{u}^{(t)} \rangle) \leq - \sum_{t=1}^{T} \langle \boldsymbol{x}^*, \boldsymbol{u}^{(t)} \rangle + \frac{\mathcal{R}(\boldsymbol{x}^*)}{\eta}$$

$$- \frac{1}{\eta} \sum_{t=1}^{T} \left( \omega(\| \boldsymbol{x}^{(t)} - \boldsymbol{g}^{(t)} \|_{\boldsymbol{x}^{(t)}}) + \omega(\| \boldsymbol{x}^{(t)} - \boldsymbol{g}^{(t-1)} \|_{\boldsymbol{g}^{(t-1)}}) \right), \tag{7}$$

---

[4]By convention, it is assumed that a sum over an empty set is 0.

for any $\boldsymbol{x}^* \in \text{int}(\mathcal{X})$. We will prove the claim for $T+1$. Indeed, applying (7) for $\boldsymbol{x}^* = \boldsymbol{g}^{(T)}$ and adding on both sides the term $\langle \boldsymbol{g}^{(T+1)} - \boldsymbol{x}^{(T+1)}, \boldsymbol{m}^{(T+1)} \rangle - \langle \boldsymbol{g}^{(T+1)}, \boldsymbol{u}^{(T+1)} \rangle$ yields that

$$
\sum_{t=1}^{T+1} \left( \langle \boldsymbol{g}^{(t)} - \boldsymbol{x}^{(t)}, \boldsymbol{m}^{(t)} \rangle - \langle \boldsymbol{g}^{(t)}, \boldsymbol{u}^{(t)} \rangle \right) \leq
$$
$$
- \left\langle \boldsymbol{g}^{(T)}, \sum_{t=1}^{T} \boldsymbol{u}^{(t)} \right\rangle + \frac{\mathcal{R}(\boldsymbol{g}^{(T)})}{\eta} + \langle \boldsymbol{g}^{(T+1)} - \boldsymbol{x}^{(T+1)}, \boldsymbol{m}^{(T+1)} \rangle - \langle \boldsymbol{g}^{(T+1)}, \boldsymbol{u}^{(T+1)} \rangle
$$
$$
- \frac{1}{\eta} \sum_{t=1}^{T} \left( \omega(\|\boldsymbol{x}^{(t)} - \boldsymbol{g}^{(t)}\|_{\boldsymbol{x}^{(t)}}) + \omega(\|\boldsymbol{x}^{(t)} - \boldsymbol{g}^{(t-1)}\|_{\boldsymbol{g}^{(t-1)}}) \right). \tag{8}
$$

Now, by the first-order optimality condition of the optimization problem associated with (BTL), we have that $\nabla \Psi^{(T)}(\boldsymbol{g}^{(T)}) = \boldsymbol{0}$. As a result, Lemma A.3 implies that

$$
-\Psi^{(T)}(\boldsymbol{x}^{(T+1)}) + \Psi^{(T)}(\boldsymbol{g}^{(T)}) \geq \omega(\|\boldsymbol{x}^{(T+1)} - \boldsymbol{g}^{(T)}\|_{\boldsymbol{g}^{(T)}}) \implies
$$
$$
- \left\langle \boldsymbol{x}^{(T+1)}, \sum_{t=1}^{T} \boldsymbol{u}^{(t)} \right\rangle + \frac{\mathcal{R}(\boldsymbol{x}^{(T+1)})}{\eta} + \left\langle \boldsymbol{g}^{(T)}, \sum_{t=1}^{T} \boldsymbol{u}^{(t)} \right\rangle - \frac{\mathcal{R}(\boldsymbol{g}^{(T)})}{\eta} \geq \frac{1}{\eta} \omega(\|\boldsymbol{x}^{(T+1)} - \boldsymbol{g}^{(T)}\|_{\boldsymbol{g}^{(T)}}), \tag{9}
$$

where we used the fact that $-\Psi^{(T)}$ is a self-concordant function, which in turn follows directly from the fact that linear perturbations do not affect self-concordance. Thus, plugging (9) to (8) yields that

$$
\sum_{t=1}^{T+1} \left( \langle \boldsymbol{g}^{(t)} - \boldsymbol{x}^{(t)}, \boldsymbol{m}^{(t)} \rangle - \langle \boldsymbol{g}^{(t)}, \boldsymbol{u}^{(t)} \rangle \right) \leq
$$
$$
- \left\langle \boldsymbol{x}^{(T+1)}, \sum_{t=1}^{T} \boldsymbol{u}^{(t)} \right\rangle + \frac{\mathcal{R}(\boldsymbol{x}^{(T+1)})}{\eta} + \langle \boldsymbol{g}^{(T+1)} - \boldsymbol{x}^{(T+1)}, \boldsymbol{m}^{(T+1)} \rangle - \langle \boldsymbol{g}^{(T+1)}, \boldsymbol{u}^{(T+1)} \rangle
$$
$$
- \frac{1}{\eta} \sum_{t=1}^{T} \left( \omega(\|\boldsymbol{x}^{(t)} - \boldsymbol{g}^{(t)}\|_{\boldsymbol{x}^{(t)}}) + \omega(\|\boldsymbol{x}^{(t)} - \boldsymbol{g}^{(t-1)}\|_{\boldsymbol{g}^{(t-1)}}) \right) - \frac{1}{\eta} \omega(\|\boldsymbol{x}^{(T+1)} - \boldsymbol{g}^{(T)}\|_{\boldsymbol{g}^{(T)}})
$$
$$
= - \left\langle \boldsymbol{x}^{(T+1)}, \boldsymbol{m}^{(T+1)} + \sum_{t=1}^{T} \boldsymbol{u}^{(t)} \right\rangle + \frac{\mathcal{R}(\boldsymbol{x}^{(T+1)})}{\eta} + \langle \boldsymbol{g}^{(T+1)}, \boldsymbol{m}^{(T+1)} \rangle - \langle \boldsymbol{g}^{(T+1)}, \boldsymbol{u}^{(T+1)} \rangle
$$
$$
- \frac{1}{\eta} \sum_{t=1}^{T} \left( \omega(\|\boldsymbol{x}^{(t)} - \boldsymbol{g}^{(t)}\|_{\boldsymbol{x}^{(t)}}) + \omega(\|\boldsymbol{x}^{(t)} - \boldsymbol{g}^{(t-1)}\|_{\boldsymbol{g}^{(t-1)}}) \right) - \frac{1}{\eta} \omega(\|\boldsymbol{x}^{(T+1)} - \boldsymbol{g}^{(T)}\|_{\boldsymbol{g}^{(T)}}). \tag{10}
$$

Similarly, by the first-order optimality condition of the optimization problem associated with (OFTRL), we have that $\nabla \Phi^{(T+1)}(\boldsymbol{x}^{(T+1)}) = \boldsymbol{0}$. Thus, by Lemma A.3 it follows that

$$
-\Phi^{(T+1)}(\boldsymbol{g}^{(T+1)}) + \Phi^{(T+1)}(\boldsymbol{x}^{(T+1)}) \geq \omega(\|\boldsymbol{x}^{(T+1)} - \boldsymbol{g}^{(T+1)}\|_{\boldsymbol{x}^{(T+1)}}),
$$

since $-\Phi^{(T+1)}$ is self-concordant. Plugging this bound to (10) implies that

$$\sum_{t=1}^{T+1} \left( \langle \boldsymbol{g}^{(t)} - \boldsymbol{x}^{(t)}, \boldsymbol{m}^{(t)} \rangle - \langle \boldsymbol{g}^{(t)}, \boldsymbol{u}^{(t)} \rangle \right) \leq$$

$$- \left\langle \boldsymbol{g}^{(T+1)}, \boldsymbol{m}^{(T+1)} + \sum_{t=1}^{T} \boldsymbol{u}^{(t)} \right\rangle + \frac{\mathcal{R}(\boldsymbol{g}^{(T+1)})}{\eta} + \langle \boldsymbol{g}^{(T+1)}, \boldsymbol{m}^{(T+1)} \rangle - \langle \boldsymbol{g}^{(T+1)}, \boldsymbol{u}^{(T+1)} \rangle$$

$$- \frac{1}{\eta} \sum_{t=1}^{T+1} \left( \omega(\|\boldsymbol{x}^{(t)} - \boldsymbol{g}^{(t)}\|_{\boldsymbol{x}^{(t)}}) + \omega(\|\boldsymbol{x}^{(t)} - \boldsymbol{g}^{(t-1)}\|_{\boldsymbol{g}^{(t-1)}}) \right)$$

$$= - \left\langle \boldsymbol{g}^{(T+1)}, \sum_{t=1}^{T+1} \boldsymbol{u}^{(t)} \right\rangle + \frac{\mathcal{R}(\boldsymbol{g}^{(T+1)})}{\eta}$$

$$- \frac{1}{\eta} \sum_{t=1}^{T+1} \left( \omega(\|\boldsymbol{x}^{(t)} - \boldsymbol{g}^{(t)}\|_{\boldsymbol{x}^{(t)}}) + \omega(\|\boldsymbol{x}^{(t)} - \boldsymbol{g}^{(t-1)}\|_{\boldsymbol{g}^{(t-1)}}) \right)$$

$$\leq - \left\langle \boldsymbol{x}^*, \sum_{t=1}^{T+1} \boldsymbol{u}^{(t)} \right\rangle + \frac{\mathcal{R}(\boldsymbol{x}^*)}{\eta} - \frac{1}{\eta} \sum_{t=1}^{T+1} \left( \omega(\|\boldsymbol{x}^{(t)} - \boldsymbol{g}^{(t)}\|_{\boldsymbol{x}^{(t)}}) + \omega(\|\boldsymbol{x}^{(t)} - \boldsymbol{g}^{(t-1)}\|_{\boldsymbol{g}^{(t-1)}}) \right),$$

for any $\boldsymbol{x}^* \in \mathrm{int}(\mathcal{X})$, where the last inequality follows since $\Psi^{(T+1)}(\boldsymbol{g}^{(T+1)}) \geq \Psi^{(T+1)}(\boldsymbol{x}^*)$, for any $\boldsymbol{x}^* \in \mathrm{int}(\mathcal{X})$, by definition of $\boldsymbol{g}^{(T+1)}$. This establishes the inductive step, completing the proof of the theorem. $\qquad \square$

Next, to cast Theorem B.1 in the form of an RVU bound (in the sense of [Syrgkanis et al., 2015]), we establish the stability of the iterates as formalized below.

**Lemma B.2** (Stability). *Let $\eta > 0$ be such that $\eta\|\boldsymbol{u}^{(t)} - \boldsymbol{m}^{(t)}\|_{*,\boldsymbol{x}^{(t)}} \leq \frac{1}{2}$ and $\eta\|\boldsymbol{m}^{(t)}\|_{*,\boldsymbol{g}^{(t-1)}} \leq \frac{1}{2}$, for all $t \in [\![T]\!]$. Then, for any $t \in [\![T]\!]$,*

$$\|\boldsymbol{x}^{(t)} - \boldsymbol{g}^{(t)}\|_{\boldsymbol{x}^{(t)}} \leq 2\eta\|\boldsymbol{u}^{(t)} - \boldsymbol{m}^{(t)}\|_{*,\boldsymbol{x}^{(t)}};$$
$$\|\boldsymbol{x}^{(t)} - \boldsymbol{g}^{(t-1)}\|_{\boldsymbol{g}^{(t-1)}} \leq 2\eta\|\boldsymbol{m}^{(t)}\|_{*,\boldsymbol{g}^{(t-1)}}.$$

*Proof.* Fix any $t \in [\![T]\!]$. We observe that $\|\boldsymbol{x}^{(t)} - \boldsymbol{g}^{(t)}\|_{\boldsymbol{x}^{(t)}} = \|\boldsymbol{x}^{(t)} - \arg\min(-\Psi^{(t)})\|_{\boldsymbol{x}^{(t)}}$, by definition of (BTL). Further, we have that $\Psi^{(t)}(\boldsymbol{x}) = \Phi^{(t)}(\boldsymbol{x}) + \eta\langle \boldsymbol{x}, \boldsymbol{u}^{(t)} - \boldsymbol{m}^{(t)} \rangle$, implying that $\nabla\Psi^{(t)} = \nabla\Phi^{(t)} + \eta(\boldsymbol{u}^{(t)} - \boldsymbol{m}^{(t)})$. By the first-order optimaility condition of the optimization problem associated with (OFTRL), it follows that $\nabla\Phi^{(t)}(\boldsymbol{x}^{(t)}) = 0$, in turn implying that $\nabla\Psi^{(t)}(\boldsymbol{x}^{(t)}) = \eta(\boldsymbol{u}^{(t)} - \boldsymbol{m}^{(t)})$. As a result, we have shown that $\lambda(\boldsymbol{x}^{(t)}, -\Psi^{(t)}) = \|\nabla\Psi^{(t)}(\boldsymbol{x}^{(t)})\|_{*,\boldsymbol{x}^{(t)}} = \eta\|\boldsymbol{u}^{(t)} - \boldsymbol{m}^{(t)}\|_{*,\boldsymbol{x}^{(t)}} \leq \frac{1}{2}$, by assumption. Thus, Lemma A.6 implies that

$$\|\boldsymbol{x}^{(t)} - \boldsymbol{g}^{(t)}\|_{\boldsymbol{x}^{(t)}} = \|\boldsymbol{x}^{(t)} - \arg\min(-\Psi^{(t)})\|_{\boldsymbol{x}^{(t)}} \leq 2\lambda(\boldsymbol{x}^{(t)}, -\Psi^{(t)}) = 2\eta\|\boldsymbol{u}^{(t)} - \boldsymbol{m}^{(t)}\|_{*,\boldsymbol{x}^{(t)}},$$

concluding the first part of the claim. Similarly, we have that $\|\boldsymbol{x}^{(t)} - \boldsymbol{g}^{(t-1)}\|_{\boldsymbol{g}^{(t-1)}} = \|\boldsymbol{g}^{(t-1)} - \arg\min(-\Phi^{(t)})\|_{\boldsymbol{g}^{(t-1)}}$, by definition of (OFTRL). Further, we observe that $\Phi^{(t)}(\boldsymbol{x}) = \Psi^{(t-1)}(\boldsymbol{x}) + \eta\langle \boldsymbol{x}, \boldsymbol{m}^{(t)} \rangle$, implying that $\nabla\Phi^{(t)} = \nabla\Psi^{(t-1)} + \eta\boldsymbol{m}^{(t)}$. Moreover, by the first-order optimality condition of the optimization problem associated with (BTL), we have that $\nabla\Psi^{(t-1)}(\boldsymbol{g}^{(t-1)}) = \boldsymbol{0}$. In turn, this implies that $\nabla\Phi^{(t)}(\boldsymbol{g}^{(t-1)}) = \eta\boldsymbol{m}^{(t)}$. As a result, we have shown that $\lambda(\boldsymbol{g}^{(t-1)}, -\Phi^{(t)}) = \|\nabla\Phi^{(t)}(\boldsymbol{g}^{(t-1)})\|_{*,\boldsymbol{g}^{(t-1)}} = \eta\|\boldsymbol{m}^{(t)}\|_{*,\boldsymbol{g}^{(t-1)}} \leq \frac{1}{2}$, by assumption. Thus, Lemma A.6 implies that

$$\|\boldsymbol{x}^{(t)} - \boldsymbol{g}^{(t-1)}\|_{\boldsymbol{g}^{(t-1)}} = \|\boldsymbol{g}^{(t-1)} - \arg\min(-\Phi^{(t)})\|_{\boldsymbol{g}^{(t-1)}} \leq 2\lambda(\boldsymbol{g}^{(t-1)}, -\Phi^{(t)}) = 2\eta\|\boldsymbol{m}^{(t)}\|_{*,\boldsymbol{g}^{(t-1)}}.$$

$\qquad \square$

We are now ready to establish Theorem 3.1, the statement of which is recalled below.

**Theorem 3.1** (RVU for Self-Concordant Regularizers). *Suppose that $\mathcal{R}$ is a nondegenerate self-concordant function for $\mathrm{int}(\mathcal{X})$. Moreover, let $\eta > 0$ be such that $\eta\|\boldsymbol{u}^{(t)} - \boldsymbol{m}^{(t)}\|_{*,\boldsymbol{x}^{(t)}} \le \frac{1}{2}$ and $\eta\|\boldsymbol{m}^{(t)}\|_{*,\boldsymbol{g}^{(t-1)}} \le \frac{1}{2}$ for all $t \in [\![T]\!]$. Then, the regret $\mathrm{Reg}^T(\boldsymbol{x}^*)$ of (OFTRL) with respect to any comparator $\boldsymbol{x}^* \in \mathrm{int}(\mathcal{X})$ under any sequence of utilities $\boldsymbol{u}^{(1)}, \ldots, \boldsymbol{u}^{(T)}$ can be bounded by*

$$\frac{\mathcal{R}(\boldsymbol{x}^*)}{\eta} + 2\eta \sum_{t=1}^{T} \|\boldsymbol{u}^{(t)} - \boldsymbol{m}^{(t)}\|_{*,\boldsymbol{x}^{(t)}}^2 - \frac{1}{4\eta} \sum_{t=1}^{T} \left( \|\boldsymbol{x}^{(t)} - \boldsymbol{g}^{(t)}\|_{\boldsymbol{x}^{(t)}}^2 + \|\boldsymbol{x}^{(t)} - \boldsymbol{g}^{(t-1)}\|_{\boldsymbol{g}^{(t-1)}}^2 \right).$$

*Proof.* First, combining Theorem B.1 with the fact that $\|\boldsymbol{x}^{(t)} - \boldsymbol{g}^{(t)}\|_{\boldsymbol{x}^{(t)}} \le 2\eta\|\boldsymbol{u}^{(t)} - \boldsymbol{m}^{(t)}\|_{*,\boldsymbol{x}^{(t)}}$ (by Lemma B.2) yields that $\mathrm{Reg}^T(\boldsymbol{x}^*)$ is upper bounded by

$$\frac{\mathcal{R}(\boldsymbol{x}^*)}{\eta} + 2\eta \sum_{t=1}^{T} \|\boldsymbol{u}^{(t)} - \boldsymbol{m}^{(t)}\|_{*,\boldsymbol{x}^{(t)}}^2 - \frac{1}{\eta} \sum_{t=1}^{T} \left( \omega(\|\boldsymbol{x}^{(t)} - \boldsymbol{g}^{(t)}\|_{\boldsymbol{x}^{(t)}}) + \omega(\|\boldsymbol{x}^{(t)} - \boldsymbol{g}^{(t-1)}\|_{\boldsymbol{g}^{(t-1)}}) \right).$$

Further, it follows from Lemma B.2 that $\|\boldsymbol{x}^{(t)} - \boldsymbol{g}^{(t)}\|_{\boldsymbol{x}^{(t)}} \le 1$ and $\|\boldsymbol{x}^{(t)} - \boldsymbol{g}^{(t-1)}\|_{\boldsymbol{g}^{(t-1)}} \le 1$. Thus, Fact A.4 implies that $\mathrm{Reg}^T(\boldsymbol{x}^*)$ is upper bounded by

$$\frac{\mathcal{R}(\boldsymbol{x}^*)}{\eta} + 2\eta \sum_{t=1}^{T} \|\boldsymbol{u}^{(t)} - \boldsymbol{m}^{(t)}\|_{*,\boldsymbol{x}^{(t)}}^2 - \frac{1}{4\eta} \sum_{t=1}^{T} \left( \|\boldsymbol{x}^{(t)} - \boldsymbol{g}^{(t)}\|_{\boldsymbol{x}^{(t)}}^2 + \|\boldsymbol{x}^{(t)} - \boldsymbol{g}^{(t-1)}\|_{\boldsymbol{g}^{(t-1)}}^2 \right).$$

$\square$

For our purposes, it will be convenient to cast Theorem 3.1 in the following form, using the additional assumption that the Hessian $\nabla^2 \mathcal{R}$ is stable.

**Corollary B.3.** *Suppose that $\mathcal{R}$ is a nondegenerate self-concordant function for $\mathrm{int}(\mathcal{X})$ such that $\nabla^2 \mathcal{R}(\widetilde{\boldsymbol{x}}) \preceq 2\nabla^2 \mathcal{R}(\boldsymbol{x})$ for any $\boldsymbol{x}, \widetilde{\boldsymbol{x}} \in \mathrm{int}(\mathcal{X})$ with $\|\boldsymbol{x} - \widetilde{\boldsymbol{x}}\|_{\widetilde{\boldsymbol{x}}} \le \frac{1}{4}$. Moreover, let $\eta > 0$ be such that $\eta\|\boldsymbol{u}^{(t)} - \boldsymbol{m}^{(t)}\|_{*,\boldsymbol{x}^{(t)}} \le \frac{1}{8}$ and $\eta\|\boldsymbol{m}^{(t)}\|_{*,\boldsymbol{g}^{(t-1)}} \le \frac{1}{2}$ for all $t \in [\![T]\!]$. Then, the regret of (OFTRL) under any sequence of utilities $\boldsymbol{u}^{(1)}, \ldots, \boldsymbol{u}^{(T)}$ can be bounded as*

$$\mathrm{Reg}^T(\boldsymbol{x}^*) \le \frac{\mathcal{R}(\boldsymbol{x}^*)}{\eta} + 2\eta \sum_{t=1}^{T} \|\boldsymbol{u}^{(t)} - \boldsymbol{m}^{(t)}\|_{*,\boldsymbol{x}^{(t)}}^2 - \frac{1}{16\eta} \sum_{t=1}^{T} \|\boldsymbol{x}^{(t)} - \boldsymbol{x}^{(t-1)}\|_{\boldsymbol{x}^{(t-1)}}^2.$$

*Proof.* First, by Lemma B.2 we know that $\|\boldsymbol{x}^{(t-1)} - \boldsymbol{g}^{(t-1)}\|_{\boldsymbol{x}^{(t-1)}} \le 2\eta\|\boldsymbol{u}^{(t-1)} - \boldsymbol{m}^{(t-1)}\|_{*,\boldsymbol{x}^{(t-1)}} \le \frac{1}{4}$, for any $t \in \mathbb{N}$. Thus, by assumption, it follows that $\nabla^2 \mathcal{R}(\boldsymbol{x}^{(t-1)}) \preceq 2\nabla^2 \mathcal{R}(\boldsymbol{g}^{(t-1)})$, in turn implying that $\|\boldsymbol{x}^{(t)} - \boldsymbol{g}^{(t-1)}\|_{\boldsymbol{x}^{(t-1)}}^2 \le 2\|\boldsymbol{x}^{(t)} - \boldsymbol{g}^{(t-1)}\|_{\boldsymbol{g}^{(t-1)}}^2$. Further, the triangle inequality for the norm $\|\cdot\|_{\boldsymbol{x}^{(t-1)}}$ implies that

$$\|\boldsymbol{x}^{(t)} - \boldsymbol{x}^{(t-1)}\|_{\boldsymbol{x}^{(t-1)}}^2 \le 2\|\boldsymbol{x}^{(t)} - \boldsymbol{g}^{(t-1)}\|_{\boldsymbol{x}^{(t-1)}}^2 + 2\|\boldsymbol{g}^{(t-1)} - \boldsymbol{x}^{(t-1)}\|_{\boldsymbol{x}^{(t-1)}}^2$$
$$\le 4\|\boldsymbol{x}^{(t)} - \boldsymbol{g}^{(t-1)}\|_{\boldsymbol{g}^{(t-1)}}^2 + 4\|\boldsymbol{x}^{(t-1)} - \boldsymbol{g}^{(t-1)}\|_{\boldsymbol{x}^{(t-1)}}^2,$$

where we used Young's inequality in the first line, and the fact that $\|\boldsymbol{x}^{(t)} - \boldsymbol{g}^{(t-1)}\|_{\boldsymbol{x}^{(t-1)}}^2 \le 2\|\boldsymbol{x}^{(t)} - \boldsymbol{g}^{(t-1)}\|_{\boldsymbol{g}^{(t-1)}}^2$ in the second line. Thus, summing over all $t \in [\![T]\!]$ yields that

$$\sum_{t=1}^{T} \|\boldsymbol{x}^{(t)} - \boldsymbol{x}^{(t-1)}\|_{\boldsymbol{x}^{(t-1)}}^2 \le 4\sum_{t=1}^{T} \|\boldsymbol{x}^{(t)} - \boldsymbol{g}^{(t-1)}\|_{\boldsymbol{g}^{(t-1)}}^2 + 4\sum_{t=1}^{T} \|\boldsymbol{x}^{(t-1)} - \boldsymbol{g}^{(t-1)}\|_{\boldsymbol{x}^{(t-1)}}^2$$
$$\le 4\sum_{t=1}^{T} \|\boldsymbol{x}^{(t)} - \boldsymbol{g}^{(t-1)}\|_{\boldsymbol{g}^{(t-1)}}^2 + 4\sum_{t=1}^{T} \|\boldsymbol{x}^{(t)} - \boldsymbol{g}^{(t)}\|_{\boldsymbol{x}^{(t)}}^2,$$

since $\boldsymbol{x}^{(0)} = \boldsymbol{g}^{(0)}$. Finally, plugging this bound to Theorem 3.1 concludes the proof. $\square$

**Corollary B.4** (Stability of the Iterates). *Suppose that $\mathcal{R}$ is a self-concordant function for $\mathrm{int}(\mathcal{X})$ such that $\nabla^2 \mathcal{R}(\widetilde{\boldsymbol{x}}) \preceq 2\nabla^2 \mathcal{R}(\boldsymbol{x})$ for any $\boldsymbol{x}, \widetilde{\boldsymbol{x}} \in \mathrm{int}(\mathcal{X})$ with $\|\boldsymbol{x} - \widetilde{\boldsymbol{x}}\|_{\widetilde{\boldsymbol{x}}} \le \frac{1}{4}$. Moreover, let $\eta > 0$ be such that $\eta\|\boldsymbol{u}^{(t)} - \boldsymbol{m}^{(t)}\|_{*,\boldsymbol{x}^{(t)}} \le \frac{1}{8}$ and $\eta\|\boldsymbol{m}^{(t)}\|_{*,\boldsymbol{g}^{(t-1)}} \le \frac{1}{2}$ for all $t \in [\![T]\!]$. Then,*

$$\|\boldsymbol{x}^{(t)} - \boldsymbol{x}^{(t-1)}\|_{\boldsymbol{x}^{(t-1)}} \le 4\eta\|\boldsymbol{m}^{(t)}\|_{*,\boldsymbol{g}^{(t-1)}} + 2\eta\|\boldsymbol{u}^{(t-1)} - \boldsymbol{m}^{(t-1)}\|_{*,\boldsymbol{x}^{(t-1)}}.$$

*Proof.* Similarly to the proof of Corollary B.3, we obtain that

$$\|\boldsymbol{x}^{(t)} - \boldsymbol{x}^{(t-1)}\|_{\boldsymbol{x}^{(t-1)}} \le \|\boldsymbol{x}^{(t)} - \boldsymbol{g}^{(t-1)}\|_{\boldsymbol{x}^{(t-1)}} + \|\boldsymbol{g}^{(t-1)} - \boldsymbol{x}^{(t-1)}\|_{\boldsymbol{x}^{(t-1)}}$$

$$\le 2\|\boldsymbol{x}^{(t)} - \boldsymbol{g}^{(t-1)}\|_{\boldsymbol{g}^{(t-1)}} + \|\boldsymbol{x}^{(t-1)} - \boldsymbol{g}^{(t-1)}\|_{\boldsymbol{x}^{(t-1)}}$$

$$\le 4\eta\|\boldsymbol{m}^{(t)}\|_{*,\boldsymbol{g}^{(t-1)}} + 2\eta\|\boldsymbol{u}^{(t-1)} - \boldsymbol{m}^{(t-1)}\|_{*,\boldsymbol{x}^{(t-1)}}.$$

$\square$

## B.1 Log-Barrier Regularizer on the Simplex

Next, we instantiate our general RVU bound for the probability simplex. To this end, let us first point out that, leveraging Lemma A.8 (and Lemma A.2), we can construct a self-concordant barrier for any polytope defined by a set of inequalities $\mathbf{A}\boldsymbol{x} \ge \boldsymbol{b}$, for a matrix $\mathbf{A} \in \mathbb{R}^{k \times d}$ and a vector $\boldsymbol{b} \in \mathbb{R}^k$, as pointed out below.

**Definition B.5** (Log-Barrier Regularizer for Polytopes)**.** Consider any polytope defined by a set of inequalities $\mathbf{A}\boldsymbol{x} \ge \boldsymbol{b}$, for a matrix $\mathbf{A} \in \mathbb{R}^{k \times d}$ and a vector $\boldsymbol{b} \in \mathbb{R}^k$. The *log-barrier* function $\mathcal{R}$ is defined as

$$\mathcal{R}(\boldsymbol{x}) := -\sum_{r=1}^{k} \log(\mathbf{A}[r,:]\boldsymbol{x} - \boldsymbol{b}[r]). \tag{11}$$

Indeed, Lemma A.8 implies that $\mathcal{R}$ is a $k$-self-concordant barrier as it can be expressed as the sum of $k$ 1-self-concordant barriers. Now let us focus on constructing a self-concordant barrier for the $(d-1)$-dimensional simplex $\Delta^d := \left\{ \boldsymbol{x} \in \mathbb{R}_{\ge 0}^d : \sum_{r=1}^{d} \boldsymbol{x}[r] = 1 \right\}$. To address the fact that $\Delta^d$ has empty interior, we will restrict the problem to the domain $\Delta^\circ := \left\{ \boldsymbol{x} \in \mathbb{R}_{\ge 0}^{d-1} : \sum_{r=1}^{d-1} \boldsymbol{x}[r] \le 1 \right\}$. For notational convenience, we will also let $\boldsymbol{x}[d] = 1 - \sum_{r=1}^{d-1} \boldsymbol{x}[r]$. Thus, using the general *log-barrier* regularizer for polytopes given in (11), we arrive at the log-barrier regularizer for $\Delta^\circ$:

$$\mathcal{R}(\boldsymbol{x}) := -\sum_{r=1}^{d-1} \log(\boldsymbol{x}[r]) - \log\left(1 - \sum_{r=1}^{d-1} \boldsymbol{x}[r]\right). \tag{12}$$

Naturally, $\mathcal{R}$ is a $d$-self-concordant barrier since it can be expressed as the sum of $d$ 1-self-concordant barriers. It is important to stress that the regularizer given in (12) takes as input a $(d-1)$-dimensional vector. To reconcile this with the fact that the regret minimizer should receive a $d$-dimensional utility vector $\boldsymbol{u} \in \mathbb{R}^d$, in the sequel we will use a simple transformation of the observed utilities (while preserving the incurred regret). But first, let us also introduce an auxiliary regularizer for the purpose of our analysis; namely,

$$\widetilde{\mathcal{R}}(\boldsymbol{x}) := -\sum_{r=1}^{d} \log \boldsymbol{x}[r]. \tag{13}$$

We are going to relate the local norm induced by the log-barrier (12) to that induced by the auxiliary regularizer (13). First, we characterize the primal local norm induced by $\mathcal{R}$ and $\widetilde{\mathcal{R}}$.

**Claim B.6.** *For any* $\boldsymbol{x}, \widetilde{\boldsymbol{x}} \in \text{int}(\Delta^\circ)$,

$$\|\boldsymbol{x} - \widetilde{\boldsymbol{x}}\|_{\widetilde{\mathcal{R}},\boldsymbol{x}}^2 = \sum_{r=1}^{d} \left(\frac{\boldsymbol{x}[r] - \widetilde{\boldsymbol{x}}[r]}{\boldsymbol{x}[r]}\right)^2.$$

*Proof.* Let us first compute the Hessian of $\mathcal{R}$. A direct calculation gives that for $r \in [\![d-1]\!]$,

$$\frac{\partial^2 \mathcal{R}}{\partial \boldsymbol{x}[r]^2} = \frac{1}{(\boldsymbol{x}[r])^2} + \frac{1}{\left(1 - \sum_{r=1}^{d-1} \boldsymbol{x}[r]\right)^2} = \frac{1}{(\boldsymbol{x}[r])^2} + \frac{1}{(\boldsymbol{x}[d])^2},$$

where recall that $\boldsymbol{x}[d] = 1 - \sum_{r=1}^{d-1} \boldsymbol{x}[r]$ (by convention). Further, for $r' \ne r \in [\![d-1]\!]$ we have that

$$\frac{\partial^2 \mathcal{R}}{\partial \boldsymbol{x}[r]\partial \boldsymbol{x}[r']} = \frac{\partial^2 \mathcal{R}}{\partial \boldsymbol{x}[r']\partial \boldsymbol{x}[r]} = \frac{1}{(\boldsymbol{x}[d])^2}.$$

Thus, the Hessian of $\mathcal{R}$ reads

$$\nabla^2 \mathcal{R} = \operatorname{diag}\left(\frac{1}{(\boldsymbol{x}[1])^2}, \ldots, \frac{1}{(\boldsymbol{x}[d-1])^2}\right) + \frac{1}{(\boldsymbol{x}[d])^2}\mathbf{1}_{d-1}\mathbf{1}_{d-1}^\top. \tag{14}$$

As a result,

$$\|\boldsymbol{x} - \widetilde{\boldsymbol{x}}\|_{\mathcal{R},\boldsymbol{x}}^2 = (\boldsymbol{x} - \widetilde{\boldsymbol{x}})^\top \operatorname{diag}\left(\frac{1}{(\boldsymbol{x}[1])^2}, \ldots, \frac{1}{(\boldsymbol{x}[d-1])^2}\right)(\boldsymbol{x} - \widetilde{\boldsymbol{x}}) + \frac{(\mathbf{1}_{d-1}^\top(\boldsymbol{x} - \widetilde{\boldsymbol{x}}))^2}{(\boldsymbol{x}[d])^2}$$

$$= \sum_{r=1}^{d-1}\left(\frac{\boldsymbol{x}[r] - \widetilde{\boldsymbol{x}}[r]}{\boldsymbol{x}[r]}\right)^2 + \left(\frac{\sum_{r=1}^{d-1}\boldsymbol{x}[r] - \sum_{r=1}^{d-1}\widetilde{\boldsymbol{x}}[r]}{\boldsymbol{x}[d]}\right)^2$$

$$= \sum_{r=1}^{d-1}\left(\frac{\boldsymbol{x}[r] - \widetilde{\boldsymbol{x}}[r]}{\boldsymbol{x}[r]}\right)^2 + \left(\frac{\boldsymbol{x}[d] - \widetilde{\boldsymbol{x}}[d]}{\boldsymbol{x}[d]}\right)^2$$

$$= \sum_{r=1}^{d}\left(\frac{\boldsymbol{x}[r] - \widetilde{\boldsymbol{x}}[r]}{\boldsymbol{x}[r]}\right)^2.$$

$\square$

Next, we characterize the dual norm induced by the regularizer $\mathcal{R}$. To this end, let us first explain how the regret minimizer over the domain $\Delta^\circ$ should operate. Upon observing a utility vector $\boldsymbol{u} \in \mathbb{R}^d$, we construct the vector $\widetilde{\boldsymbol{u}} \in \mathbb{R}^{d-1}$ so that $\widetilde{\boldsymbol{u}}[r] = \boldsymbol{u}[r] - \boldsymbol{u}[d]$, for all $r \in [\![d-1]\!]$. It is easy to see that the regret incurred is preserved through this transformation.

**Claim B.7.** *For any $\widetilde{\boldsymbol{u}} \in \mathbb{R}^{d-1}$ and $\boldsymbol{x} \in \operatorname{int}(\Delta^\circ)$,*

$$\|\widetilde{\boldsymbol{u}}\|_{*,\mathcal{R},\boldsymbol{x}} = \|\boldsymbol{u} - c^*\mathbf{1}_d\|_{*,\widetilde{\mathcal{R}},\boldsymbol{x}},$$

*where $c^*$ is the scalar that minimizes the norm in the right hand side.*

*Proof.* First, using the Sherman–Morrison formula we find that the inverse of the Hessian of $\mathcal{R}$ given in (14) can be expressed as

$$(\nabla^2\mathcal{R})^{-1} = \operatorname{diag}(\widetilde{\boldsymbol{x}}[1], \ldots, \widetilde{\boldsymbol{x}}[d-1]) - \frac{1}{\sum_{r=1}^d(\boldsymbol{x}[r])^2}\widetilde{\boldsymbol{x}}\widetilde{\boldsymbol{x}}^\top,$$

where $\widetilde{\boldsymbol{x}} \coloneqq ((\boldsymbol{x}[1])^2, \ldots, (\boldsymbol{x}[d-1])^2)$. Thus, by definition of $\widetilde{\boldsymbol{u}}$ we have that

$$\|\widetilde{\boldsymbol{u}}\|_{*,\mathcal{R},\boldsymbol{x}} = \sum_{r=1}^{d-1}(\boldsymbol{x}[r])^2(\boldsymbol{u}[r] - \boldsymbol{u}[d])^2 - \frac{(\sum_{r=1}^{d-1}(\boldsymbol{x}[r])^2(\boldsymbol{u}[r] - \boldsymbol{u}[d]))^2}{\sum_{r=1}^d(\boldsymbol{x}[r])^2}$$

$$= \sum_{r=1}^{d}(\boldsymbol{x}[r])^2(\boldsymbol{u}[r] - \boldsymbol{u}[d])^2 - \frac{(\sum_{r=1}^{d}(\boldsymbol{x}[r])^2\boldsymbol{u}[r] - \boldsymbol{u}[d]\sum_{r=1}^{d}(\boldsymbol{x}[r])^2)^2}{\sum_{r=1}^d(\boldsymbol{x}[r])^2}$$

$$= \sum_{r=1}^{d}(\boldsymbol{x}[r])^2(\boldsymbol{u}[r] - \boldsymbol{u}[d])^2 - \frac{(\sum_{r=1}^{d}(\boldsymbol{x}[r])^2\boldsymbol{u}[r])^2}{\sum_{r=1}^d(\boldsymbol{x}[r])^2}$$

$$+ 2\boldsymbol{u}[d]\sum_{r=1}^{d}(\boldsymbol{x}[r])^2\boldsymbol{u}[r] - (\boldsymbol{u}[d])^2\sum_{r=1}^{d}(\boldsymbol{x}[r])^2$$

$$= \sum_{r=1}^{d}(\boldsymbol{x}[r]\boldsymbol{u}[r])^2 - \frac{(\sum_{r=1}^{d}(\boldsymbol{x}[r])^2\boldsymbol{u}[r])^2}{\sum_{r=1}^d(\boldsymbol{x}[r])^2}, \tag{15}$$

by simple algebraic calculations. Now let us define the scalar $c^*$ as

$$c^* \coloneqq \frac{\sum_{r=1}^{d}(\boldsymbol{x}[r])^2\boldsymbol{u}[r]}{\sum_{r=1}^{d}(\boldsymbol{x}[r])^2}.$$

Then, continuing from (15),

$$\|\widetilde{\boldsymbol{u}}\|_{*,\mathcal{R},\boldsymbol{x}} = \sum_{r=1}^{d} (\boldsymbol{x}[r])^2 \left( (\boldsymbol{u}[r])^2 - 2 \left( \frac{\sum_{r'=1}^{d}(\boldsymbol{x}[r'])^2\boldsymbol{u}[r']}{\sum_{r'=1}^{d}(\boldsymbol{x}[r'])^2} \right) \boldsymbol{u}[r] + \left( \frac{\sum_{r'=1}^{d}(\boldsymbol{x}[r'])^2\boldsymbol{u}[r']}{\sum_{r'=1}^{d}(\boldsymbol{x}[r'])^2} \right)^2 \right)$$

$$= \sum_{r=1}^{r} (\boldsymbol{x}[r])^2 \left( \boldsymbol{u}[r] - c^* \right)^2 = \|\boldsymbol{u} - c^*\boldsymbol{1}_d\|_{*,\widetilde{\mathcal{R}},\boldsymbol{x}}. \tag{16}$$

But, it is easy to see that $c^*$ is the minimizer of (16). This concludes the proof. $\qquad\square$

An analogous argument shows that $\|\widetilde{\boldsymbol{u}}^{(t)} - \widetilde{\boldsymbol{u}}^{(t-1)}\|_{*,\mathcal{R},\boldsymbol{x}} = \|\boldsymbol{u}^{(t)} - \boldsymbol{u}^{(t-1)} - c^*\boldsymbol{1}_d\|_{*,\widetilde{\mathcal{R}},\boldsymbol{x}} \leq \|\boldsymbol{u}^{(t)} - \boldsymbol{u}^{(t-1)}\|_{*,\widetilde{\mathcal{R}},\boldsymbol{x}}$. Finally, combining Claim B.6 and Claim B.7 with Theorem 3.1 and Corollary B.3 directly leads to the RVU bound of Corollary 3.2.

## C  Omitted Proofs from Section 4

In this section we provide the omitted proofs from Section 4. We start with the proof of Lemma 4.2. To this end, we first apply Corollary 3.2 for each individual regret minimizer $\mathfrak{R}_a$, leading to the following guarantee for $\eta \leq \frac{1}{16}$.

$$\mathrm{Reg}_a^T(\boldsymbol{x}_a^*) \leq \frac{\mathcal{R}(\boldsymbol{x}_a^*)}{\eta} + 2\eta \sum_{t=1}^{T} \|\boldsymbol{u}^{(t)}\boldsymbol{x}^{(t)}[a] - \boldsymbol{u}^{(t-1)}\boldsymbol{x}^{(t-1)}[a]\|^2_{*,\boldsymbol{x}_a^{(t)}}$$

$$- \frac{1}{16\eta} \sum_{t=1}^{T} \|\boldsymbol{x}_a^{(t)} - \boldsymbol{x}_a^{(t-1)}\|^2_{\boldsymbol{x}_a^{(t-1)}}, \tag{17}$$

for any $\boldsymbol{x}_a^* \in \mathrm{relint}(\Delta(\mathcal{A}))$; it is assumed that each regret minimizer $\mathfrak{R}_a$ is employing the same learning rate $\eta > 0$. Next, the triangle inequality along with Young's inequality imply that

$$\|\boldsymbol{u}^{(t)}\boldsymbol{x}^{(t)}[a] - \boldsymbol{u}^{(t-1)}\boldsymbol{x}^{(t-1)}[a]\|^2_{*,\boldsymbol{x}_a^{(t)}} \leq 2(\boldsymbol{x}^{(t)}[a])^2\|\boldsymbol{u}^{(t)} - \boldsymbol{u}^{(t-1)}\|^2_{*,\boldsymbol{x}_a^{(t)}}$$

$$+ 2(\boldsymbol{x}^{(t)}[a] - \boldsymbol{x}^{(t-1)}[a])^2\|\boldsymbol{u}^{(t-1)}\|^2_{*,\boldsymbol{x}_a^{(t)}},$$

for any $a \in \mathcal{A}$. Summing this inequality over all $a \in \mathcal{A}$ yields that

$$\sum_{a \in \mathcal{A}} \|\boldsymbol{u}^{(t)}\boldsymbol{x}^{(t)}[a] - \boldsymbol{u}^{(t-1)}\boldsymbol{x}^{(t-1)}[a]\|^2_{*,\boldsymbol{x}_a^{(t)}} \leq 2\|\boldsymbol{u}^{(t)} - \boldsymbol{u}^{(t-1)}\|_\infty^2 + 2\|\boldsymbol{x}^{(t)} - \boldsymbol{x}^{(t-1)}\|_2^2. \tag{18}$$

Next, let us address the diameter term in (17). Let $\boldsymbol{x}_c := \arg\min_{\boldsymbol{x}} \mathcal{R}(\boldsymbol{x})$, so that $\mathcal{R}(\boldsymbol{x}_c) = 0$. If $\pi(\boldsymbol{x}_a^*; \boldsymbol{x}_c) \leq 1 - \frac{1}{T}$, then, by Theorem A.9,

$$\mathcal{R}(\boldsymbol{x}_a^*) \leq |\mathcal{A}| \log \left( \frac{1}{1 - \pi(\boldsymbol{x}_a^*; \boldsymbol{x}_c)} \right) \leq m \log T,$$

where we used the notation $m := |\mathcal{A}|$. Otherwise, we define $\widetilde{\boldsymbol{x}}_a^* := (1 - 1/T)\boldsymbol{x}_a^* + (1/T)\boldsymbol{x}_c$, and we observe that

$$\mathrm{Reg}_a^T(\boldsymbol{x}_a^*) \leq \mathrm{Reg}_a^T(\widetilde{\boldsymbol{x}}_a^*) + \sum_{t=1}^{T} \langle \boldsymbol{x}_a^* - \widetilde{\boldsymbol{x}}_a^*, \boldsymbol{x}^{(t)}[a]\boldsymbol{u}^{(t)} \rangle \leq \mathrm{Reg}_a^T(\widetilde{\boldsymbol{x}}_a^*) + \frac{2}{T} \sum_{t=1}^{T} \boldsymbol{x}^{(t)}[a]\|\boldsymbol{u}^{(t)}\|_\infty.$$

Thus, from (17) we conclude that

$$\mathrm{Reg}_a^T \leq \frac{m \log T}{\eta} + \frac{2}{T} \sum_{t=1}^{T} \boldsymbol{x}^{(t)}[a] + 2\eta \sum_{t=1}^{T} \|\boldsymbol{u}^{(t)}\boldsymbol{x}^{(t)}[a] - \boldsymbol{u}^{(t-1)}\boldsymbol{x}^{(t-1)}[a]\|^2_{*,\boldsymbol{x}_a^{(t)}}$$

$$- \frac{1}{16\eta} \sum_{t=1}^{T} \|\boldsymbol{x}_a^{(t)} - \boldsymbol{x}_a^{(t-1)}\|^2_{\boldsymbol{x}_a^{(t-1)}}, \tag{19}$$

since $\|\boldsymbol{u}^{(t)}\|_\infty \leq 1$. Next, we will use the fact that the log-barrier regularizer guarantees *multiplicative stability*, in the following formal sense.

**Corollary C.1** (Multiplicative Stability)**.** *In the setting of Corollary B.4, suppose that* $\|\boldsymbol{u}^{(t)}\|_\infty, \|\boldsymbol{m}^{(t)}\|_\infty \leq 1$ *for all* $t \in \llbracket T \rrbracket$. *If* $\eta \leq \frac{1}{16}$, *then for* $2 \leq t \leq T$,

$$\sqrt{\sum_{r=1}^d \left(1 - \frac{\boldsymbol{x}^{(t)}[r]}{\boldsymbol{x}^{(t-1)}[r]}\right)^2} \leq 6\eta\|\boldsymbol{u}^{(t-1)}\|_\infty + 2\eta\|\boldsymbol{u}^{(t-2)}\|_\infty.$$

*Proof.* The claim follows directly from Corollary B.4 with $\boldsymbol{m}^{(t)} = \boldsymbol{u}^{(t-1)}$, using the fact that $\|\boldsymbol{u}\|_{*,\boldsymbol{x}^{(t)}} \leq \|\boldsymbol{u}\|_\infty$. $\qquad\square$

Now let

$$\mu_a^{(t)} := \max_{a' \in \mathcal{A}} \left|1 - \frac{\boldsymbol{x}_a^{(t)}[a']}{\boldsymbol{x}_a^{(t-1)}[a']}\right|,$$

for each $a \in \mathcal{A}$. Corollary C.1 implies that

$$\mu_a^{(t)} \leq 6\eta\|\boldsymbol{u}^{(t-1)}\boldsymbol{x}^{(t-1)}[a]\|_\infty + 2\eta\|\boldsymbol{u}^{(t-2)}\boldsymbol{x}^{(t-2)}[a]\|_\infty$$
$$= 6\eta\boldsymbol{x}^{(t-1)}[a]\|\boldsymbol{u}^{(t-1)}\|_\infty + 2\eta\boldsymbol{x}^{(t-2)}[a]\|\boldsymbol{u}^{(t-2)}\|_\infty.$$

Thus, summing over all $a \in \mathcal{A}$ yields that

$$\sum_{a \in \mathcal{A}} \mu_a^{(t)} \leq 6\eta \sum_{a \in \mathcal{A}} \boldsymbol{x}^{(t-1)}[a]\|\boldsymbol{u}^{(t-1)}\|_\infty + 2\eta \sum_{a \in \mathcal{A}} \boldsymbol{x}^{(t-2)}[a]\|\boldsymbol{u}^{(t-2)}\|_\infty \leq 8\eta, \qquad (20)$$

for $t \geq 2$, where we used that $\boldsymbol{x}^{(t-1)}, \boldsymbol{x}^{(t-2)} \in \Delta(\mathcal{A})$, as well as the normalization assumption $\|\boldsymbol{u}\|_\infty \leq 1$; it is also immediate to see that $\sum_{a \in \mathcal{A}} \mu_a^{(1)} \leq 8\eta$.

For the proof of Lemma 4.2 we will require the Markov chain tree theorem. In particular, consider an $m$-node ergodic (*i.e.*, aperiodic and irreducible) Markov chain represented through a row-stochastic matrix $\mathbf{Q}$. The Markov chain tree theorem establishes a closed-form solution for the (unique) stationary distribution $\boldsymbol{\pi}$; that is, the vector $\boldsymbol{\pi} \in \Delta^m$ for which $\boldsymbol{\pi}^\top \mathbf{Q} = \boldsymbol{\pi}^\top$. To this end, we formalize the notion of a directed tree.

**Definition C.2** (Directed Tree)**.** A directed graph $\mathcal{T} = (V, E)$ is a *directed tree* rooted at node $a$ if (i) it containts no (directed) cycles; (ii) every node $V \setminus \{a\}$ has exactly one outgoing edge; and (iii) the root node $a$ has no outgoing edges.

We will denote with $\mathbb{T}_a$ the set of all possible directed $m$-node trees rooted at node $a$. Finally, before we state the Markov chain tree theorem, we let $\Sigma_a$ be defined as

$$\Sigma_a = \sum_{\mathcal{T} \in \mathbb{T}_a} \prod_{(u,v) \in E(\mathcal{T})} \mathbf{Q}[u, v]. \qquad (21)$$

**Theorem C.3** (Markov Chain Tree Theorem; *e.g.*, [Anantharam and Tsoucas, 1989])**.** *The stationary distribution* $\boldsymbol{\pi} \in \Delta^m$ *of an* $m$*-state ergodic markov chain with row-stochastic transition matrix* $\mathbf{Q}$ *is such that*

$$\boldsymbol{\pi}[a] = \frac{\Sigma_a}{\Sigma},$$

*where* $\Sigma := \sum_a \Sigma_a$, *and each* $\Sigma_a$ *is defined as in* (21).

We are now ready to prove Lemma 4.2.

**Lemma 4.2.** *Suppose that each regret minimizer* $\mathfrak{R}_a$ *employs* (OFTRL) *with log-barrier regularization and* $\eta \leq \frac{1}{16}$. *Then, for any* $t \in \mathbb{N}$,

$$\|\boldsymbol{x}^{(t)} - \boldsymbol{x}^{(t-1)}\|_1^2 \leq 64|\mathcal{A}| \sum_{a \in \mathcal{A}} \|\boldsymbol{x}_a^{(t)} - \boldsymbol{x}_a^{(t-1)}\|_{\boldsymbol{x}_a^{(t-1)}}^2.$$

*Proof.* Consider any $t \in \mathbb{N}$. From the Markov chain tree theorem (Theorem C.3) we know that

$$\boldsymbol{x}[a] = \frac{\Sigma_a}{\Sigma}, \quad \forall a \in \mathcal{A},$$

where $\Sigma_a := \sum_{\mathcal{T} \in \mathbb{T}_a} \prod_{(u,v) \in E(\mathcal{T})} \mathbf{Q}[u,v]$ and $\Sigma = \sum_{a \in \mathcal{A}} \Sigma_a$. Fix some action $a \in \mathcal{A}$ and a directed tree $\mathcal{T} \in \mathbb{T}_a$ rooted at node $a$. Then,

$$\prod_{(u,v) \in E(\mathcal{T})} \mathbf{Q}^{(t)}[u,v] = \prod_{(u,v) \in E(\mathcal{T})} \boldsymbol{x}_u^{(t)}[v] \leq \prod_{(u,v) \in E(\mathcal{T})} (1 + \mu_u^{(t)}) \boldsymbol{x}_u^{(t-1)}[v],$$

where we used the fact that

$$\mu_u^{(t)} \geq \frac{\boldsymbol{x}_u^{(t)}[v]}{\boldsymbol{x}_u^{(t-1)}[v]} - 1 \implies \boldsymbol{x}_u^{(t)}[v] \leq (1 + \mu_u^{(t)}) \boldsymbol{x}_u^{(t-1)}[v],$$

Thus,

$$\prod_{(u,v) \in E(\mathcal{T})} \mathbf{Q}^{(t)}[u,v] \leq \prod_{u \neq a} (1 + \mu_u^{(t)}) \prod_{(u,v) \in E(\mathcal{T})} \mathbf{Q}^{(t-1)}[u,v],$$

where we used the fact that $\mathcal{T}$ is a directed tree rooted at $a$. Thus, summing over all $\mathcal{T} \in \mathbb{T}_a$ yields that

$$\Sigma_a^{(t)} = \sum_{\mathcal{T} \in \mathbb{T}_a} \prod_{(u,v) \in E(\mathcal{T})} \mathbf{Q}^{(t)}[u,v] \leq \Sigma_a^{(t-1)} \prod_{a' \in \mathcal{A}} (1 + \mu_{a'}^{(t)})$$

$$\leq \Sigma_a^{(t-1)} \exp \left\{ \sum_{a' \in \mathcal{A}} \mu_{a'}^{(t)} \right\}. \tag{22}$$

This also implies that

$$\Sigma^{(t)} = \sum_{a \in \mathcal{A}} \Sigma_a^{(t)} \leq \exp \left\{ \sum_{a' \in \mathcal{A}} \mu_{a'}^{(t)} \right\} \sum_{a \in \mathcal{A}} \Sigma_a^{(t-1)} = \exp \left\{ \sum_{a' \in \mathcal{A}} \mu_{a'}^{(t)} \right\} \Sigma^{(t-1)}. \tag{23}$$

Similarly,

$$\prod_{(u,v) \in E(\mathcal{T})} \mathbf{Q}^{(t)}[u,v] = \prod_{(u,v) \in E(\mathcal{T})} \boldsymbol{x}_u^{(t)}[v] \geq \prod_{(u,v) \in E(\mathcal{T})} (1 - \mu_u^{(t)}) \boldsymbol{x}_u^{(t-1)}[v],$$

where we used the fact that

$$\mu_u^{(t)} \geq 1 - \frac{\boldsymbol{x}_u^{(t)}[v]}{\boldsymbol{x}_u^{(t-1)}[v]} \implies \boldsymbol{x}_u^{(t)}[v] \geq (1 - \mu_u^{(t)}) \boldsymbol{x}_u^{(t-1)}[v].$$

Thus, summing over all $\mathcal{T} \in \mathbb{T}_a$ implies that

$$\Sigma_a^{(t)} = \sum_{\mathcal{T} \in \mathbb{T}_a} \prod_{(u,v) \in E(\mathcal{T})} \mathbf{Q}^{(t)}[u,v] \geq \Sigma_a^{(t-1)} \prod_{a' \in \mathcal{A}} (1 - \mu_{a'}^{(t)})$$

$$\geq \Sigma_a^{(t-1)} \exp \left\{ -2 \sum_{a' \in \mathcal{A}} \mu_{a'}^{(t)} \right\}, \tag{24}$$

where we used the inequality $1 - x \geq e^{-2x}$, for all $x \in [0, \frac{1}{2}]$, applicable since (by (20)) $\sum_{a' \in \mathcal{A}} \mu_{a'}^{(t)} \leq 8\eta \leq \frac{1}{2}$ for $\eta \leq \frac{1}{16}$. This also implies that

$$\Sigma^{(t)} = \sum_{a \in \mathcal{A}} \Sigma_a^{(t)} \geq \exp \left\{ -2 \sum_{a' \in \mathcal{A}} \mu_{a'}^{(t)} \right\} \sum_{a \in \mathcal{A}} \Sigma_a^{(t-1)} = \exp \left\{ -2 \sum_{a' \in \mathcal{A}} \mu_{a'}^{(t)} \right\} \Sigma^{(t-1)}. \tag{25}$$

As a result, from (22) and (25) it follows that for any $a \in \mathcal{A}$,

$$\frac{\Sigma_a^{(t)}}{\Sigma^{(t)}} - \frac{\Sigma_a^{(t-1)}}{\Sigma^{(t-1)}} \leq \frac{\Sigma_a^{(t-1)} \exp \left\{ \sum_{a' \in \mathcal{A}} \mu_{a'}^{(t)} \right\}}{\Sigma^{(t-1)} \exp \left\{ -2 \sum_{a' \in \mathcal{A}} \mu_{a'}^{(t)} \right\}} - \frac{\Sigma_a^{(t-1)}}{\Sigma^{(t-1)}} = \frac{\Sigma_a^{(t-1)}}{\Sigma^{(t-1)}} \left( \exp \left\{ 3 \sum_{a' \in \mathcal{A}} \mu_{a'}^{(t)} \right\} - 1 \right)$$

$$\leq \frac{\Sigma_a^{(t-1)}}{\Sigma^{(t-1)}} \left( 8 \sum_{a' \in \mathcal{A}} \mu_{a'}^{(t)} \right),$$

where we used the inequality $e^x - 1 \leq \frac{8}{3}x$ for all $x \in [0, \frac{3}{2}]$, applicable since $\sum_{a' \in \mathcal{A}} \mu_{a'}^{(t)} \leq \frac{1}{2}$. Similarly, (24) and (23) imply that for any $a \in \mathcal{A}$,

$$\frac{\Sigma_a^{(t-1)}}{\Sigma^{(t-1)}} - \frac{\Sigma_a^{(t)}}{\Sigma^{(t)}} \leq \frac{\Sigma_a^{(t-1)}}{\Sigma^{(t-1)}} - \frac{\Sigma_a^{(t-1)} \exp\left\{-2 \sum_{a' \in \mathcal{A}} \mu_{a'}^{(t)}\right\}}{\Sigma^{(t-1)} \exp\left\{\sum_{a' \in \mathcal{A}} \mu_{a'}^{(t)}\right\}} = \frac{\Sigma_a^{(t-1)}}{\Sigma^{(t-1)}}\left(1 - \exp\left\{-3 \sum_{a' \in \mathcal{A}} \mu_{a'}^{(t)}\right\}\right)$$

$$\leq \frac{\Sigma_a^{(t-1)}}{\Sigma^{(t-1)}}\left(3 \sum_{a' \in \mathcal{A}} \mu_{a'}^{(t)}\right).$$

As a result, we have established that

$$\left|\boldsymbol{x}^{(t)}[a] - \boldsymbol{x}^{(t-1)}[a]\right| = \left|\frac{\Sigma_a^{(t)}}{\Sigma^{(t)}} - \frac{\Sigma_a^{(t-1)}}{\Sigma^{(t-1)}}\right| \leq 8\frac{\Sigma_a^{(t-1)}}{\Sigma^{(t-1)}} \sum_{a' \in \mathcal{A}} \mu_{a'}^{(t)} = 8\boldsymbol{x}^{(t-1)}[a] \sum_{a' \in \mathcal{A}} \mu_{a'}^{(t)},$$

in turn implying that

$$\|\boldsymbol{x}^{(t)} - \boldsymbol{x}^{(t-1)}\|_1 \leq 8\left(\sum_{a' \in \mathcal{A}} \mu_{a'}^{(t)}\right)\left(\sum_{a \in \mathcal{A}} \boldsymbol{x}^{(t-1)}[a]\right) = 8 \sum_{a' \in \mathcal{A}} \mu_{a'}^{(t)}, \tag{26}$$

since $\boldsymbol{x}^{(t-1)} \in \Delta(\mathcal{A})$. Thus,

$$\|\boldsymbol{x}^{(t)} - \boldsymbol{x}^{(t-1)}\|_1^2 \leq 64\left(\sum_{a \in \mathcal{A}} \mu_a^{(t)}\right)^2 \leq 64|\mathcal{A}| \sum_{a \in \mathcal{A}}\left(\mu_a^{(t)}\right)^2,$$

by Jensen's inequality. Finally,

$$\left(\mu_a^{(t)}\right)^2 = \max_{a' \in \mathcal{A}}\left(1 - \frac{\boldsymbol{x}_a^{(t)}[a']}{\boldsymbol{x}_a^{(t-1)}[a']}\right)^2 \leq \sum_{a' \in \mathcal{A}}\left(1 - \frac{\boldsymbol{x}_a^{(t)}[a']}{\boldsymbol{x}_a^{(t-1)}[a']}\right)^2 = \|\boldsymbol{x}_a^{(t)} - \boldsymbol{x}_a^{(t-1)}\|_{\boldsymbol{x}_a^{(t-1)}}^2,$$

and combining this bound with (26) concludes the proof. $\square$

**Theorem 4.3** (RVU Bound for Swap Regret). *Suppose that each $\mathfrak{R}_a$ employs* (OFTRL) *with log-barrier regularization and $\eta \leq \frac{1}{128\sqrt{m}}$. Then, for $T \geq 2$, the swap regret of $\mathfrak{R}_{swap}$ is bounded as*

$$\text{SwapReg}^T \leq \frac{2m^2 \log T}{\eta} + 4\eta \sum_{t=1}^{T} \|\boldsymbol{u}^{(t)} - \boldsymbol{u}^{(t-1)}\|_\infty^2 - \frac{1}{2048m\eta} \sum_{t=1}^{T} \|\boldsymbol{x}^{(t)} - \boldsymbol{x}^{(t-1)}\|_1^2.$$

*Proof.* Combining (19), (18), Theorem 4.1, and Lemma 4.2 implies that $\text{SwapReg}_i^T$ is upper bounded by

$$\frac{2m^2 \log T}{\eta} + 4\eta \sum_{t=1}^{T} \|\boldsymbol{u}^{(t)} - \boldsymbol{u}^{(t-1)}\|_\infty^2 + 4\eta \sum_{t=1}^{T} \|\boldsymbol{x}^{(t)} - \boldsymbol{x}^{(t-1)}\|_2^2 - \frac{1}{1024m\eta} \sum_{t=1}^{T} \|\boldsymbol{x}^{(t)} - \boldsymbol{x}^{(t-1)}\|_1^2.$$

Further, for $\eta \leq \frac{1}{128\sqrt{m}}$ it follows that

$$4\eta\|\boldsymbol{x}^{(t)} - \boldsymbol{x}^{(t-1)}\|_2^2 \leq 4\eta\|\boldsymbol{x}^{(t)} - \boldsymbol{x}^{(t-1)}\|_1^2 \leq \frac{1}{2048m\eta}\|\boldsymbol{x}^{(t)} - \boldsymbol{x}^{(t-1)}\|_1^2.$$

In turn, this implies that

$$\text{SwapReg}_i^T \leq \frac{2m^2 \log T}{\eta} + 4\eta \sum_{t=1}^{T} \|\boldsymbol{u}^{(t)} - \boldsymbol{u}^{(t-1)}\|_\infty^2 - \frac{1}{2048m\eta} \sum_{t=1}^{T} \|\boldsymbol{x}^{(t)} - \boldsymbol{x}^{(t-1)}\|_1^2,$$

concluding the proof. $\square$

**Corollary 4.5** (Near-Optimal Individual Swap Regret). *Suppose that all players use* BM-OFTRL-LogBar *with* $\eta = \frac{1}{128(n-1)\max_{i\in[\![n]\!]}\{\sqrt{m_i}\}}$. *Then, the individual swap regret* $\mathrm{SwapReg}_i^T$ *up to time $T \geq 2$ of each player $i \in [\![n]\!]$ can be bounded as*

$$\mathrm{SwapReg}_i^T \leq 256 \max_{j\in[\![n]\!]}\{\sqrt{m_j}\} \left( (n-1)m_i^2 + \sum_{j=1}^{n} m_j^2 \right) \log T.$$

*Proof.* By Theorem 4.3 and Theorem 4.4,

$$\mathrm{SwapReg}_i^T \leq \frac{2m_i^2 \log T}{\eta} + 4\eta(n-1) \sum_{j\neq i} \sum_{t=1}^{T} \|\boldsymbol{x}_j^{(t)} - \boldsymbol{x}_j^{(t-1)}\|_1^2$$

$$\leq \frac{2m_i^2 \log T}{\eta} + 32768\eta(n-1) \max_{j\in[\![n]\!]}\{m_j\} \sum_{j=1}^{n} m_j^2 \log T$$

$$= 256 \max_{j\in[\![n]\!]}\{\sqrt{m_j}\} \left( (n-1)m_i^2 + \sum_{j=1}^{n} m_j^2 \right) \log T.$$

$\square$

**Corollary 4.6** (Adversarial Robustness). *There exist dynamics such that when all players follow them the individual swap regret of each player grows as in Corollary 4.5. Moreover, when faced against adversarial utilities, such that $\|\boldsymbol{u}_i^{(t)}\|_\infty \leq 1$ for all $t \in [\![T]\!]$, the algorithm guarantees that*

$$\mathrm{SwapReg}_i^T \leq 256 \max_{j\in[\![n]\!]}\{\sqrt{m_j}\} \left( (n-1)m_i^2 + \sum_{j=1}^{n} m_j^2 \right) \log T + 2\sqrt{m_i \log m_i T} + 2.$$

*Proof.* Each player $i \in [\![n]\!]$ initially follows the BM-OFTRL-LogBar dynamics with learning rate $\eta = \frac{1}{128(n-1)\max_{j\in[\![n]\!]}\{\sqrt{m_j}\}}$. Next, player $i$ keeps track of the quantity $\sum_{\tau=1}^{t} \|\boldsymbol{u}_i^{(\tau)} - \boldsymbol{u}_i^{(\tau-1)}\|_\infty^2$. If for all $2 \leq t \leq T$ it holds that

$$\sum_{\tau=1}^{t} \|\boldsymbol{u}_i^{(\tau)} - \boldsymbol{u}_i^{(\tau-1)}\|_\infty^2 \leq 8192(n-1) \max_{j\in[\![n]\!]}\{m_j\} \sum_{j=1}^{n} m_j^2 \log t, \qquad (27)$$

then the swap regret of player $i \in [\![n]\!]$ enjoys the guarantee of Corollary 4.5, as follows directly from Theorem 4.3. In particular, (27) will hold as long as all players follow the prescribed dynamics, by virtue of Theorem 4.4. Otherwise, let $t \geq 2$ be the *first* iteration for which (27) is violated. The overall swap regret accumulated up to time $t - 1$ is at most the guarantee of Corollary 4.5, as follows directly from Theorem 4.3, while the swap regret at time $t$ is at most 2 since $\|\boldsymbol{u}_i^{(t)}\|_\infty \leq 1$. Next, the player switches to BM-MWU with learning rate $\eta = \sqrt{\frac{m_i \log m_i}{T}}$. Thereafter, the accumulated swap regret will be bounded by $2\sqrt{m_i \log m_i T}$. This completes the proof. $\square$

Finally, we conclude this section with a refinement for games with a large number of players. In particular, we will assume that the utility of each player only depends on the actions of a small number of other players (Item 1), and that each player's actions only affect the utility of a small number of other players (Item 2). Understanding whether the linear dependence of Corollary 4.5 on $n$ is necessary in general games is left as an interesting open question.

**Theorem C.4** (Refinement for Large Games). *Suppose that all players use* BM-OFTRL-LogBar. *Furthermore, assume that the utility of player $i \in [\![n]\!]$ depends on a subset of players $\mathcal{N}_i \subseteq [\![n]\!]$, so that*

1. $|\mathcal{N}_i| \leq c \leq n - 1$; *and*

2. $\max_{i\in[\![n]\!]} |\{j \neq i : i \in \mathcal{N}_j\}| \leq c$.

*Then, for $\eta = \frac{1}{128c\max_{j\in[\![n]\!]}\{\sqrt{m_j}\}}$,*

$$\sum_{i=1}^{n} \mathrm{SwapReg}_i^T \le 256c \max_{j\in[\![n]\!]}\{\sqrt{m_j}\} \sum_{j=1}^{n} m_j^2 \log T.$$

*Moreover, for $\eta = \frac{1}{128\sqrt{cn}\max_{j\in[\![n]\!]}\{\sqrt{m_j}\}} \le \frac{1}{128c\max_{j\in[\![n]\!]}\{\sqrt{m_j}\}}$,*

$$\mathrm{SwapReg}_i^T \le 256 \max_{j\in[\![n]\!]}\{\sqrt{m_j}\} \left( \sqrt{cn}m_i^2 + \sqrt{\frac{c}{n}} \sum_{j=1}^{n} m_j^2 \right) \log T.$$

*In particular, if $m_i = m$ for all $i \in [\![n]\!]$,*

$$\mathrm{SwapReg}_i^T \le 512\sqrt{cn}m^{5/2} \log T.$$

*Proof.* The proof proceeds similarly to the proof of Theorem 4.4. First, we have that

$$\left( \|\boldsymbol{u}_i^{(t)} - \boldsymbol{u}_i^{(t-1)}\|_\infty \right)^2 \le \left( \sum_{j\in\mathcal{N}_i} \|\boldsymbol{x}_j^{(t)} - \boldsymbol{x}_j^{(t-1)}\|_1 \right)^2 \le c \sum_{j\ne i} \|\boldsymbol{x}_j^{(t)} - \boldsymbol{x}_j^{(t-1)}\|_1^2,$$

since $|\mathcal{N}_i| \le c$. Thus, using Theorem 4.3, $\sum_{i=1}^{n} \mathrm{SwapReg}_i^T$ can be upper bounded by

$$2\frac{\log T}{\eta} \sum_{i=1}^{n} m_i^2 + 4\eta c \sum_{i=1}^{n} \sum_{j\in\mathcal{N}_i} \sum_{t=1}^{T} \|\boldsymbol{x}_j^{(t)} - \boldsymbol{x}_j^{(t-1)}\|_1^2 - \sum_{i=1}^{n} \frac{1}{2048m_i\eta} \sum_{t=1}^{T} \|\boldsymbol{x}_i^{(t)} - \boldsymbol{x}_i^{(t-1)}\|_1^2$$

$$\le 2\frac{\log T}{\eta} \sum_{i=1}^{n} m_i^2 + \sum_{i=1}^{n} \left( 4\eta c^2 - \frac{1}{2048m_i\eta} \right) \sum_{t=1}^{T} \|\boldsymbol{x}_i^{(t)} - \boldsymbol{x}_i^{(t-1)}\|_1^2 \qquad (28)$$

$$\le 2\frac{\log T}{\eta} \sum_{i=1}^{n} m_i^2 - \frac{1}{4096\eta} \sum_{i=1}^{n} \frac{1}{m_i} \sum_{t=1}^{T} \|\boldsymbol{x}_i^{(t)} - \boldsymbol{x}_i^{(t-1)}\|_1^2, \qquad (29)$$

where (28) uses the assumption that $|\{j \ne i : i \in \mathcal{N}_j\}| \le c$, for any player $i \in [\![n]\!]$, and (29) follows since $\eta \le \frac{1}{128c\sqrt{m_i}}$ for all $i \in [\![n]\!]$. As a result, for $\eta = \frac{1}{128c\max\{\sqrt{m_j}\}}$,

$$\sum_{i=1}^{n} \mathrm{SwapReg}_i^T \le 256c \max_{j\in[\![n]\!]}\{\sqrt{m_j}\} \sum_{j=1}^{n} m_j^2 \log T.$$

Furthermore, given that $\sum_{i=1}^{n} \mathrm{SwapReg}_i^T \ge 0$,

$$\frac{1}{\max_{j\in[\![n]\!]}\{m_j\}} \sum_{i=1}^{n} \sum_{t=1}^{T} \|\boldsymbol{x}_i^{(t)} - \boldsymbol{x}_i^{(t-1)}\|_1^2 \le \sum_{i=1}^{n} \frac{1}{m_i} \sum_{t=1}^{T} \|\boldsymbol{x}_i^{(t)} - \boldsymbol{x}_i^{(t-1)}\|_1^2 \le 8192 \sum_{i=1}^{n} m_i^2 \log T.$$

Thus, for $\eta = \frac{1}{128\sqrt{cn}\max_{j\in[\![n]\!]}\{\sqrt{m_j}\}} \le \frac{1}{128c\max_{j\in[\![n]\!]}\{\sqrt{m_j}\}}$,

$$\mathrm{SwapReg}_i^T \le 256\sqrt{cn} \max_{j\in[\![n]\!]}\{\sqrt{m_j}\}m_i^2 \log T + 256\sqrt{\frac{c}{n}} \max_{j\in[\![n]\!]}\{\sqrt{m_j}\} \sum_{j=1}^{n} m_j^2 \log T.$$

$\square$

Hence, when $c$ is a small constant this theorem implies an improvement of $\Theta(n)$ for the sum of the players' swap regrets, as well as an $\Theta(\sqrt{n})$ factor for each individual swap regret.

## D Experiments

In this section we include additional experiments in order to corroborate some of our theoretical results. First, regarding Figure 1 in the main body, we considered a bimatrix (general-sum) game described with the following payoff matrices.

$$\mathbf{A} = \begin{pmatrix} 0 & 0.5 & 1.5 \\ 1.5 & 0 & 1 \\ 0.5 & 1.5 & 0 \end{pmatrix}; \qquad \mathbf{B} = \begin{pmatrix} 0 & 1.5 & 1 \\ 1 & 0 & 1.5 \\ 1.5 & 1 & 0 \end{pmatrix}. \tag{30}$$

This game is a slight variant of *Shapley's game* [Shapley, 1964], a general-sum two-player game used by Shapley in order to illustrate that fictitious play does not converge to Nash equilibria in general-sum games. Shapley's game is not suited to illustrate the cycling behavior of the dynamics in our case since it has a unique Nash equilibrium, occurring when both players play uniformly at random; as such, (OFTRL) is initialized at the equilibrium. On the other hand, the (unique) equilibrium of the game described in (30) occurs when $\boldsymbol{x}^* = (\frac{1}{3}, \frac{1}{3}, \frac{1}{3})$ and $\boldsymbol{y}^* = (\frac{1}{4}, \frac{2}{5}, \frac{7}{20})$ [Avis et al., 2010]. As illustrated in Figure 1, `BM-OFTRL-LogBar` does not appear to converge to a Nash equilibrium—at least in a *last-iterate* sense. In contrast, we conjecture that the last iterate of (OFTRL) with log-barrier regularization converges to the set of Nash equilibria in zero-sum games, and this property seems plausible even under the BM construction.

Moreover, we conduct experiments on random $3 \times 3$ bimatrix (normal-form) general-sum games. Specifically, each entry of the payoff matrices is an independent random variable drawn from the uniform distribution in $[-1, 1]$. In Figure 2 we illustrate the swap regret of the `BM-OFTRL-LogBar` algorithm with a time-invariant learning rate $\eta = 0.1$.

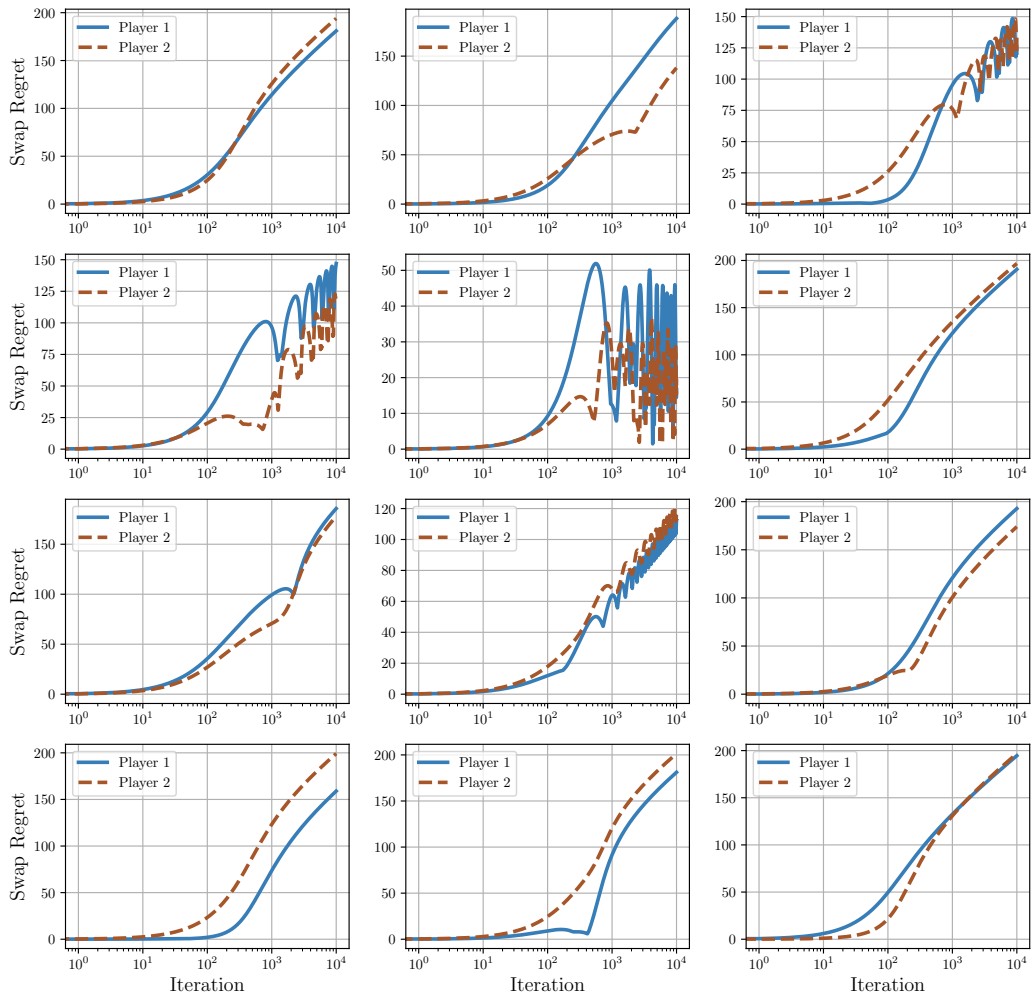

Figure 2: The swap regret experienced by each player for $T = 10^4$ iterations when both players employ `BM-OFTRL-LogBar` with $\eta = 0.1$. Each plot corresponds to a random $3 \times 3$ bimatrix game. The $x$-axis represents the iteration, in logarithmic scale, while the $y$-axis shows the swap regret experienced by each player at the given iteration. These results corroborate the $O(\log T)$ rates established in Corollary 4.5, showing that our analysis is essentially tight.