# OpenReview forum: "Uncoupled Learning Dynamics with $O(\log T)$ Swap Regret in Multiplayer Games"
_NeurIPS.cc/2022/Conference — NeurIPS 2022 Accept_

### Official Review · Reviewer_Wcat · 2022-07-10

**Rating:** 8
**Confidence:** 4
**Soundness:** 4 excellent
**Presentation:** 3 good
**Contribution:** 3 good

**Summary:**

The authors consider the problem of no-regret learning in general-sum multiplayer games. They propose an algorithm for each player which guarantees that the individual swap regret after $T$ rounds is bounded by $O(\log T)$, improving upon the previously best known bound of $O(\log^4 T)$ in general-sum games. This in turn leads to an algorithm for computing an approximate correlated equilibrium in general-sum games. The proposed algorithm for each player is decentralized, in the sense that every player only needs to observe her own losses in order to update her policy after each round, and is oblivious to the actions taken by other players. In addition, the authors establish adversarial robustness of the proposed algorithm, i.e. that each player is guaranteed $O(\sqrt{T})$ regret when faced with an adversarial loss sequence.

The algorithm follows a swap-regret minimization paradigm introduced originally by Blum and Mansour (2007), instantiated over several instances of "optimistic follow-the-regularized-leader" with a log-barrier regularizer. The analysis in the paper mainly hinges on two properties of the learning dynamics: $O(\log T)$-bounded second-order path length of each player's policy sequence, and an RVU property of the swap-regret minimization algorithm.

**Questions:**

The method of analysis presented in the paper naturally raises the question of whether or not the Optimistic Hedge algorithm (which has been shown by Daskalakis et al. [2021] to obtain polylog($T$) regret) enjoys a bounded path length property similar to the property given in Theorem 4.4 for BM-OFTRL-LogBar. If this was indeed the case, then combining the RVU property of optimistic hedge with this bounded path length would yield a logarithmic regret bound, with a considerably simpler algorithm. My question for the authors regarding this observation is whether or not they looked into the possibility of optimistic hedge having log-bounded path length, or rather is it maybe straightforward to see that it does not have this property?

**Limitations:**

See the main review.

**Strengths And Weaknesses:**

Strengths:
* The main result of this paper is highly non-trivial and strictly improves upon the previously known regret guarantees in general-sum games in terms of the dependence on $T$. In addition, this result is achieved by a simple and efficient decentralized algorithm.
* The analysis of the proposed algorithm is novel, interesting and surprising in its simplicity compared to previous works. Specifically, the use of swap-regret minimization in order to guarantee non-negativity of the regret, which together with a non-trivial RVU property, yields a path length bound on the players' iterates, and in turn the desired regret bound.
* The proofs of the technical claims presented in the paper are clear and the novel ideas in the analysis may be useful for future works.

Weaknesses:
* The following is only a minor issue, though I think it's worth mentioning.  The regret bound presented in the paper suffers from a polynomial dependence on $m_i$, the number of actions available to each player. This is in contrast to the previously best known regret bound of Daskalakis et al. (2021) in which the dependence on the number of actions is only logarithmic. Presumably this dependence on $m_i$ stems from the fact that each player $i$ uses an algorithm comprised of $m_i$ instances of optimistic FTRL, and this causes the number of actions to appear as a multiplicative factor in the regret bound. I think the authors could briefly discuss this discrepancy compared to the previous result by Daskalakis et al., even though it is arguably a lot less substantial than the improved dependence on $T$.

---

> ### Author Response · Authors · 2022-07-29
> **Response to Reviewer Wcat**
>
> We thank the reviewer for the helpful feedback. Below we address the reviewer’s questions.
>
> --- *“The following is only a minor issue, though I think it's worth mentioning. The regret bound presented in the paper suffers from a polynomial dependence on $m_i$, the number of actions available to each player. This is in contrast to the previously best known regret bound of Daskalakis et al. (2021) in which the dependence on the number of actions is only logarithmic. Presumably this dependence on $m_i$ stems from the fact that each player i uses an algorithm comprised of [...]”*
>
> While the regret bound of Daskalakis et al. (2021) depends logarithmically on the number of actions, that bound applies only to external regret. In contrast, for swap regret all the known bounds depend polynomially on the number of actions since, as the reviewer points out, each player employs one external regret minimizer for each action. This discrepancy between external and swap is, in fact, inherent as is evident by the $\text{poly}(m)$ lower bound of Blum and Mansour (2007).
>
> --- *“The method of analysis presented in the paper naturally raises the question of whether or not the Optimistic Hedge algorithm (which has been shown by Daskalakis et al. [2021] to obtain $\text{polylog}(T)$) regret enjoys a bounded path length property similar to the property given in Theorem 4.4 for BM-OFTRL-LogBar. If this was indeed the case, then combining the RVU property of optimistic hedge with this bounded path length would yield a logarithmic regret bound, with a considerably simpler algorithm. My question for the authors regarding this observation is whether or not they looked into the possibility of optimistic hedge having log-bounded path length, or rather is it maybe straightforward to see that it does not have this property?”*
>
> This is a great question, and, in our opinion, the most important open question stemming from our work. Based on some empirical evidence, it seems that Optimistic Hedge enjoys a constant second-order path length bound. Should that be proven, that would immediately imply the first $O(1)$ regret bound for each player. However, the technique we employ for obtaining an RVU bound for swap regret does not apply for Optimistic Hedge since Lemma 4.2 crucially uses the local norm induced by the log-barrier. So, analyzing Optimistic Hedge within our framework would likely require new ideas.
>
> References
>
> Blum and Mansour (2007): From external to internal regret.
>
> Daskalakis et al. (2021): Near-optimal no-regret learning in general games.

---

> > ### Comment · Reviewer_Wcat · 2022-08-07
> > **Response to authors' comment**
> >
> > I thank the authors for their response.
> >
> > --- "While the regret bound of Daskalakis et al. (2021) depends logarithmically on the number of actions..."
> >
> > Indeed I missed the fact that for swap regret, the rates must scale polynomially with the number of actions as opposed to external regret - so I retract my comment regarding this issue.
> >
> > --- "This is a great question, and, in our opinion, the most important open question stemming from our work..."
> >
> > I find the authors' comment on the possibility of Optimistic Hedge enjoying a bounded second order path length very interesting. Since they mention that it is the most important question stemming from their work, I would expect some more elaborate discussion on this manner and the relevant future research question, as I didn't find a noticeable mention of it in their paper.

---

### Official Review · Reviewer_YHrW · 2022-07-11

**Rating:** 8
**Confidence:** 3
**Soundness:** 4 excellent
**Presentation:** 4 excellent
**Contribution:** 4 excellent

**Summary:**


This paper studies how to design an uncoupled online learning algorithm to learn a correlated equilibrium, a fundamental problem in online learning and game theory. This paper develop a novel framework to analyze Optimistic FTRL that bypasses the high-order differentiation framework of [DFG21]. It gives regret bounds with better $T$ dependency compared to previous work.


**Questions:**


1. I would think it as a little overclaim by saying the framework *bypassing* the *cumbersome* framework of [DFG21]. First, [DFG21] could give regret bound that only scale with $m$ while this paper requires $m^{5/2}$. Therefore, I think it is inappropriate to say that the framework could bypass the framework of [DFG21], because it could be possible (though might be unlikely) that the framework of [DFG21] could not be bypassed to show regret bound that only scales with $m$.
3. In Table 1, $O(\sqrt{m \log mT})$ should be written as $O(\sqrt{m T \log m})$ in order to avoid confusation with $O(\sqrt{m \log(mT)})$.
4. If one only need to converge to coarse correlated equilibrium, could the algorithm in this paper (perhaps without the BM modification) give better regret bound with dependency on $m$ smaller than $m^{5/2}$?
5. Do we have a good explanation for why some plots in Figure 2 oscillates? and why the plot (2,2) seems not to scale with $O(\log T)$ at all?

**Limitations:**


1. The regret bound only scales with $m^{5/2}$, making it worse than the bounds in previous work. Though the $\log T$ result is strong, from a learning perspective, it is more important to optimize dependency on polynomial factors instead of poly-log factors. I think the author acknowledged this.

**Strengths And Weaknesses:**



Strengths:
1. This paper gives the first regret bound that only scales with $O(\log T)$, improving upon the previous best $O(\log^4 T)$ dependency.
2. The algorithm in this paper maintains both $\log T$-type swap regret and $\sqrt{T}$-type adversarial regret, which is desirable from both theoretical and empirical perspectives.
3. Synthetic experiments are conducted to illustrate the theoretical results.
4. The paper is well-organized and the writing is good. I did not find typo.

See Limitations for Weaknesses.

---

> ### Author Response · Authors · 2022-07-29
> **Response to Reviewer YHrW**
>
> We thank the reviewer for the helpful feedback. Below we address the reviewer's questions.
>
> --- *“I would think it as a little overclaim by saying the framework bypassing the cumbersome framework [...]”*
>
> First, we clarify that the analysis of Daskalakis et al. (2021) gives a logarithmic dependence on the number of actions, but their bound applies only to *external regret*. On the other hand, for *swap regret* all known results require a poly(m) dependence; note that there is in fact a $\text{poly}(m)$ lower bound for swap regret due to Blum and Mansour (2007). Now the framework of Daskalakis et al. (2021) subsequently led to a refined analysis of the algorithm of Blum and Mansour with an $O(n m^4 \log m \log^4(T))$ regret bound (see Anagnostides et al. (2021)), which we improve both in terms of the dependence on m and T. It is in that sense that we bypass their framework, but we were definitely not attempting to understate in any way the breakthrough result of Daskalakis et al. We will clarify this.
>
> --- *“In Table 1 [...]”*
>
> Thanks, we will modify the entry as we see how it can cause confusion.
>
> --- *“If one only need to converge to coarse correlated equilibrium, could the algorithm in this paper (perhaps without the BM modification) give better regret bound with dependency on [...]”*
>
> That’s a very good question. For coarse correlated equilibria, it is direct to see that our framework gives a linear dependence on $m$, instead of $m^{5/2}$ we have in the paper. The basic idea is that, from the perspective of Phi-regret, one can consider the set of deviations Phi that includes all constant transformations, along with the identity transformation; that guarantees nonnegativity for the regret. We chose to focus on swap regret because of its important implications for correlated equilibria, as well as the fact that it is the strongest notion of hindsight rationality in games. We will make sure to elaborate on these nuances in our revised version.
>
> --- *“Do we have a good explanation for why some plots in Figure 2 oscillates? and why the plot (2,2) seems not to scale with $O(\log T)$ at all?”*
>
> First, regarding the oscillations, since these experiments are performed on general-sum two-player games, we do expect (in general) oscillatory behavior; otherwise, the dynamics would converge to Nash equilibria, which in general-sum games is precluded by many impossibility results. Indeed, notice that if no-swap-regret learning dynamics converge pointwise, then the limit point has to be a Nash equilibrium—since it is a correlated equilibrium that also happens to be a product distribution.
>
> It is hard to say why the (2,2) plot does not appear to scale with $O(\log T)$, but it is not surprising that in some games the performance we observe is slightly better than the theoretical worst-case guarantee our analysis provides.
>
> --- *“The regret bound only scales with $m^{5/2}$, making it worse than the bounds in previous work.”*
>
> As we highlight in Table 1, our regret bound in terms of $m$ is actually the best known near-optimal guarantee using the fundamental algorithm of Blum and Mansour (2007). Note that Anagnostides et al. (2021) obtained only a dependence of $m^4 \log m$ in terms of $m$ for the algorithm of Blum and Mansour, although we note that they also analyzed a different algorithm, due to Stoltz and Lugosi, that led to an $m \log m$ dependence.
>
> References
>
> Blum and Mansour (2007):  From external to internal regret.
>
> Daskalakis et al. (2021): Near-optimal no-regret learning in general games.
>
> Anagnostides et al. (2021): Near-optimal no-regret learning for correlated equilibria in multi-player general-sum games.

---

### Official Review · Reviewer_8Ear · 2022-07-11

**Rating:** 8
**Confidence:** 4
**Soundness:** 4 excellent
**Presentation:** 3 good
**Contribution:** 3 good

**Summary:**

This paper studies accelerated no-regret learning algorithms for minimizing the swap regret in multi-player normal-form games. The main contribution is an algorithm that enjoys $O(\log T)$ swap regret when employed by all players simultaneously, which improves slightly over the prior best result $O(\log^4 T)$. The algorithm BM-OMWU-LogBar combines the classical Blum & Mansour’s no-swap-to-external reduction (henceforth BM reduction), and Optimistic Follow-The-Regularized-Leader (OFTRL) with log-barrier regularization for external regret minimization.


**Questions:**

In Table 1, for fair comparisons, perhaps the algorithm “SL-OMWU” from (Anagnostides et al. 2021) should also be mentioned? That algorithm achieves $O(n\log m\log ^4 T)$ internal regret, and thus $O(nm\log m\log^4 T)$ swap regret, by the standard bound SwapReg <= m * max{InternalReg, 0}.

I feel like the importance of the “nonnegativity of regret” should be emphasized even more in the presentation. Right now, this fact is only briefly mentioned in Line 93-97 & 261-262. In my opinion, this is precisely what allows the proof to be much simpler than (Daskalakis et al. 2021), and closer to the original arguments of (Rakhlin & Sridharan 2013, Syrgkanis et al. 2015).

---
After rebuttal: I thank the authors for their response, and would be glad to keep my current evaluation of the paper.


**Limitations:**

/

**Strengths And Weaknesses:**

Strengths:

This paper is an addition to the line of work on accelerated no-regret learning in normal-form games, when all players deploy the same no-regret learning algorithm. This type of results imply faster than $1/\sqrt{T}$ convergence to equilibria in games and thus could be of interest to the online learning / games community.
In particular, the result can be seen as a follow-up to the recent breakthrough of $O(\log^4 T)$ regret of Optimistic Hedge in multiplayer general-sum games (Daskalakis et al. 2021) and $O(\log^4 T)$ swap regret in the follow-up work of (Anagnostides et al. 2021).

The main strength of this paper, in my opinion, is a new proof route of this type of result using the nonnegativity of individual regrets plus the log-barrier regularization. The proof route appears to be much simpler than the above existing work.
* The “nonnegativity of individual regrets”, which holds for the swap regret but not the usual external regret, allows the individual swap regrets for each player to be bounded by their total (summed) regrets, which is much simpler to bound. For external regret, this lack of nonnegativity was precisely the reason why $O({\rm polylog}(T))$ regret in multiplayer general-sum games (Daskalakis et al. 2021) was much more challenging to establish than two-player zero-sum games (Rakhlin & Sridharan 2013).
* The log-barrier regularizer induces an RVU bound whose “negative iterate stability” term directly controls the stability of the fixed-point operation within the BM reduction (Corollary 3.2 & Lemma 4.2), which can then directly bound the "positive loss stability" term within the RVU bound when summed over all players

Overall, I think this line of arguments is new to this line of work (even though the two facts above are standard / known), and substantially simplifies the challenge of establishing $O({\rm polylog}(T))$ regret over the arguments of (Daskalakis et al. 2021, Anagnostides et al. 2021).

Weaknesses:

The improvement from $\log ^4T$ to $\log T$ is perhaps a bit incremental. There is also an improvement in the action ($m$) dependence over the prior algorithm BM-OMWU, though I think another algorithm SL-OMWU in the same work achieves even better action dependence already? (see “Questions” below.)

---

> ### Author Response · Authors · 2022-07-29
> **Response to Reviewer 8Ear**
>
> We thank the reviewer for the helpful feedback. Below we address the reviewer’s questions.
>
> --- *“In Table 1, for fair comparisons, perhaps the algorithm “SL-OMWU” from (Anagnostides et al., 2021) should also be mentioned?“*
>
> Our intention was to highlight prior results about the algorithm of Blum and Mansour (2007), but we will also include the bound regarding the algorithm of SL-OMWU in our revised version as we agree with the reviewer that it might cause some confusion.
>
> --- *“I feel like the importance of the “nonnegativity of regret” should be emphasized even more in the presentation”*
>
> We will make sure to emphasize more the importance of nonnegative regret in our revised version, as we agree with the reviewer that this simple observation is at the heart of our technique.
>
> References
>
> Blum and Mansour (2007): From external to internal regret.
>
> Anagnostides et al. (2021): Near-optimal no-regret learning for correlated equilibria in multi-player general-sum games.

---

### Meta-Review · Area_Chair_v45Z · 2022-08-24

**Recommendation:** Accept
**Confidence:** Certain

**Metareview:**

Given the unanimous support from the reviewers, to which I genuinely agree, the paper is recommended for acceptance.  I encourage the authors to pay close attention to the reviewers comments and suggestions (and in particular to the comments of Reviewer Wcat) when working on their final version.

**Award:**

No

---

### Decision · Program_Chairs · 2022-09-14

Accept